# CoMem: Compositional Concept-Graph Memory for Vision–Language Adaptation

**Heng Zhou**[1†], **Jing Tang**[2†], **Jusheng Zhang**[3],
**Yanshu Li**[4], **Canran Xiao**[5*], **Liwei Hou**[6], **Zong Ke**[1], **Jiawei Yao**[7]
[1]National University of Singapore, [2]Huazhong University of Science and Technology
[3]Sun Yat-sen University, [4]Brown University, [5]Shenzhen Campus of Sun Yat-sen University
[6]Hunan Airon Tech, [7]University of Washington
heng.zhou@u.nus.edu; xiaocr3@mail.sysu.edu.cn

## Abstract

Continual vision–language learning is crucial for multimodal tasks such as image–text retrieval, visual question answering, and grounded reasoning in dynamic environments, yet deployed systems must learn from non-stationary streams under strict privacy and memory budgets, where naïve finetuning forgets and harms transfer. We aim to sustain stable yet plastic capability in this setting without storing raw data, enabling reuse and recombination across domains and tasks. We present CoMem, a framework that treats compositional structure as the unit of memory and rehearsal: it incrementally organizes knowledge into a compact graph of concepts and relations and rehearses directly in feature space by conditioning practice signals on sampled subgraphs. A lightweight compositional consistency objective keeps part–whole predictions coherent, while teacher-informed, uncertainty-aware filtering limits off-manifold drift. Across cross-domain retrieval, structured concept learning, and continual multimodal VQA, CoMem achieves state-of-the-art retention and transfer alongside consistent gains on SVLC and VQACL/CLOVE under matched memory and parameter budgets. By casting structure as memory and rehearsing where learning happens (feature space), CoMem provides a privacy-friendly and testable paradigm for reliable continual adaptation without raw exemplars.

## 1 Introduction

Foundation vision–language models (VLMs) such as CLIP (Radford et al., 2021) have become widely used as standard backbones for a variety of multimodal tasks, including image–text retrieval (Yang et al., 2024), visual question answering (Hu et al., 2024; Xu et al., 2025; Zhang et al., 2025b), grounded reasoning (Zhu et al., 2024; Shi et al., 2025; Zhang et al., 2025d; Ma et al., 2024a), and others(Liu et al., 2025b; Ma et al., 2024b; 2025). These models enable robust cross-modal understanding by jointly embedding visual and textual information. However, in practical deployment, these systems often face challenges arising from non-stationary and domain-shifting data streams, strict privacy and memory budgets that limit the ability to retain historical samples, and heterogeneous objectives that frequently lack reliable task identifiers (Mao et al., 2022).

These challenges can lead to a significant issue: catastrophic forgetting, where fine-tuning on new tasks causes degradation in performance on previously learned tasks. This problem worsens under conditions such as domain shifts, where models struggle to maintain zero-shot performance, and with the distortion of cross-modal geometry that is crucial for transfer learning (Zheng et al., 2023; Ni et al., 2023). Studies have shown that when pretraining continues without effective retention mechanisms, forgetting tends to accumulate over time, compounding the issue and further undermining performance in real-world tasks (Garg et al., 2024).

Existing solutions typically focus on three approaches: (i) preserving cross-modal geometry and limiting parameter drift (Zheng et al., 2023; Ni et al., 2023; Zhu et al., 2023), (ii) replacing raw-

---

*Corresponding author
[†]These authors contributed equally to this work

data replay with symbolic or synthetic surrogates (Smith et al., 2023; Wu et al., 2025; Zhang et al., 2025c), and (iii) reducing the number of trainable parameters via adapters or prompts (Liu et al., 2025a). While these methods offer improvements in retention and performance, they often fail to directly address the central issue of maintaining *stable yet plastic compositional competence* when dealing with non-stationary, multi-domain data streams, especially under strict privacy and memory constraints (Zhang et al., 2021; Xiao et al., 2025). In structured-concept and skill–object tasks, models still encounter difficulties in reusing knowledge learned from earlier stages, particularly when there are shifts in the data distribution or limited supervision (Smith et al., 2023; Zhang et al., 2023; 2021). Moreover, surrogate replay methods can inherit biases from the teacher model and provide little control over what is rehearsed, while geometry-only objectives tend to preserve alignment without fostering generalization. Parameter-efficient tuning often results in task-specific adjustments, which limits the ability to reuse learned structures. As such, there is a clear need for a unified approach that combines semantically grounded rehearsal signals with mechanisms to preserve transferability across domains.

To address these gaps, we introduce COMEM, a novel continual learning framework for vision–language tasks. In COMEM, we treat *compositional structure* as the unit of memory and rehearsal. Rather than storing raw examples, COMEM incrementally organizes tasks into a compact graph of concepts and relations. Rehearsal is then conducted in feature space, conditioning practice signals on graph substructures. This method enables the model to revisit informative combinations of concepts and relations, even under strict privacy and memory constraints. Additionally, we introduce a compositional consistency objective that ensures predictions remain compatible across concepts and relations, enabling the model to reuse learned structures effectively across shifting tasks. Teacher-informed filtering and uncertainty-aware distillation mechanisms are incorporated to balance the trade-off between plasticity and stability in continual learning.

The main contributions are as follows:

1) **Structure-as-memory.** We recast continual VLM learning as organizing a compact graph of concepts and relations, then rehearse *in feature space* by conditioning on its substructures. This yields targeted, privacy-friendly practice signals without storing raw images and provides a scalable unit of reuse across tasks.

2) **Compositional stability.** We propose a training principle that maintains consistency between parts and wholes while using teacher- and uncertainty-informed filtering to balance plasticity and stability. The approach is complementary to geometry-based objectives and compatible with parameter-efficient tuning.

3) **Reliable gains under fair budgets.** On cross-domain retrieval, structured concept learning, and continual VQA, COMEM delivers higher recall and accuracy with lower forgetting under matched memory and trainable-parameter budgets, and exhibits stable behavior across seeds and reasonable hyperparameter ranges.

## 2 RELATED WORK

**Geometry and regularization.** A line of methods stabilizes CLIP-like models by constraining representation geometry or parameter drift. ZSCL preserves zero-shot transfer via unlabeled-reference distillation with weight averaging and introduces an MTIL benchmark (Zheng et al., 2023); Mod-X aligns off-diagonal similarity structure (Ni et al., 2023); CTP adds a compatible momentum branch with topology-preserving distillation (Zhu et al., 2023); DKR rectifies teacher affinities for retrieval (Cui et al., 2024). Probabilistic finetuning with language-guided consolidation (CLAP4CLIP) (Jha et al., 2024) and replay-free zero-shot stability (ZAF) (Gao et al., 2024), as well as LoRA-based consolidation (C-CLIP) and modality-gap modeling (MG-CLIP) (Liu et al., 2025a; Huang et al., 2025b), further improve stability under domain/class shift. Yet these feature/parameter-space approaches seldom model reusable concepts and typed relations, limiting compositional transfer. COMEM complements them by inducing a typed concept graph and enforcing composition via relation-aware replay and consistency.

**Replay without raw data.** Replay mitigates forgetting without storing raw data, meeting privacy and memory constraints (Zhang et al., 2025a). IncCLIP synthesizes hard negative texts with cross-

modal distillation (Yan et al., 2022); ConStruct-VL offers a data-free SVLC benchmark with adversarial pseudo-replay and Layered-LoRA (Smith et al., 2023); for VQA, SGP replays scene-graph prompts with pseudo QA pairs (Lei et al., 2023); diffusion-based synthesis (GIFT) distills on generated image–text pairs with adaptive consolidation (Wu et al., 2025). Yet symbolic or pixel-level surrogates weakly encode relations and provide limited control in the feature space where learning occurs. COMEM instead replays feature-level samples conditioned on sampled subgraphs, enabling structured, on-manifold rehearsal under tight memory budgets.

**Parameter-efficient adaptation.** Parameter-efficient adaptation (adapters, prompts, MoE) limits trainables while mitigating forgetting. TRIPLET decouples multimodal prompts for continual VQA (Qian et al., 2023), DDAS routes inputs to MoE-adapters with a frozen-CLIP fallback for OOD (Yu et al., 2024), C-CLIP couples LoRA with contrastive consolidation (Liu et al., 2025a), and CL-MoE introduces dual-router momentum experts for MLLM VQA (Huai et al., 2025). Recent advances include Proxy-FDA, which aligns neighborhood structure via proxy features (Huang et al., 2025a), and LADA, which appends label-specific memory units to a frozen encoder (Luo et al., 2025). COMEM is orthogonal to PEFT and can pair with adapters/LoRA under matched parameter or memory budgets.

## 3 METHOD

We address continual learning over a stream of multimodal tasks $\{\mathcal{D}_t\}_{t=1}^T$, where each task $t$ supplies supervision on image–text pairs $\mathcal{D}_t = \{(x_i^{(t)}, y_i^{(t)})\}_{i=1}^{n_t}$ together with task-specific labels (e.g., classes, masks). Rather than storing instances, we maintain a *concept graph memory* that encodes reusable atomic concepts and typed relations, and we perform rehearsal by generating *feature-level* samples conditioned on compositional subgraphs. As overviewed in Fig. 1, the training loop at task $t$ consists of three stages: (i) **concept induction** from $(x, y)$ with a noise-aware, teacher-frozen verifier to update the graph memory; (ii) **graph-conditioned replay** that samples a subgraph and synthesizes features from the memory; and (iii) **joint optimization** on real and synthetic batches with multi-objective regularization to preserve multimodal alignment and compositional consistency.

### 3.1 PRELIMINARIES AND NOTATION

We denote the vocabulary of attributes and entities by $\mathcal{C} = \mathcal{A} \cup \mathcal{E}$ and a set of relation types by $\mathcal{R}$. The image encoder $f_{\text{img}}(\cdot; \phi)$ maps an image to a tokenized feature map $Z = f_{\text{img}}(x; \phi) \in \mathbb{R}^{P \times d}$ with $P$ patch tokens, and the text encoder $f_{\text{txt}}(\cdot; \varphi)$ maps text to token embeddings $T = f_{\text{txt}}(y; \varphi) \in \mathbb{R}^{L \times d}$. For tasks that operate on a global descriptor, we use a pooling operator $\pi : \mathbb{R}^{P \times d} \to \mathbb{R}^d$ and write $z = \pi(Z)$. A task head $h(\cdot; \omega)$ consumes either $z$ (e.g., classification, retrieval) or $Z$ (e.g., dense prediction via a lightweight decoder). We collect trainable student parameters as $\theta = (\phi, \varphi, \omega)$ and maintain a frozen teacher $\bar{\theta}$ (the checkpoint after task $t-1$). Generator and aggregator parameters are kept separate as $\vartheta$ and $\psi$.

### 3.2 NOISE-AWARE CONCEPT INDUCTION AND VERIFICATION

For each pair $(x, y)$ we extract scored concept triplets

$$\mathcal{T}(x,y) = \Big\{ (a, e, r, w) \ : \ a \in \mathcal{A}, \ e \in \mathcal{E}, \ r \in \mathcal{R}, \ w \in [0,1] \Big\}. \tag{1}$$

Candidates $(a, e, r)$ come from a lightweight text parser $\Pi_{\text{text}}(y)$ (prompted Information Extraction, prompted IE), followed by a visual verifier evaluated on the teacher $\bar{\theta}$ to avoid confirmation bias.

Let $p(c)$ be a short prompt for concept or relation $c \in \mathcal{C} \cup \mathcal{R}$, and $t_c = f_{\text{txt}}(p(c); \bar{\varphi}) \in \mathbb{R}^d$ the *teacher* embedding. We use a *shared*, low-rank projection for verification:

$$W = AB^\top, \quad A, B \in \mathbb{R}^{d \times r}, \ r \ll d, \qquad s_{\text{align}}(c \mid Z) = \sigma\Big( \tfrac{1}{\tau} \text{LSE}_{p \in [P]} \langle W Z_p, \ t_c \rangle \Big), \tag{2}$$

where $s_{\text{align}}(c \mid Z)$ is the alignment score of concept $c$ with respect to the feature set $Z$, calculated using cosine similarity between $Z_p$ and teacher embedding $t_c$, controlled by the temperature parameter $\tau$. LSE is log-sum-exp, $\sigma$ is the sigmoid, and $\tau$ a temperature. Triplet confidence aggregates

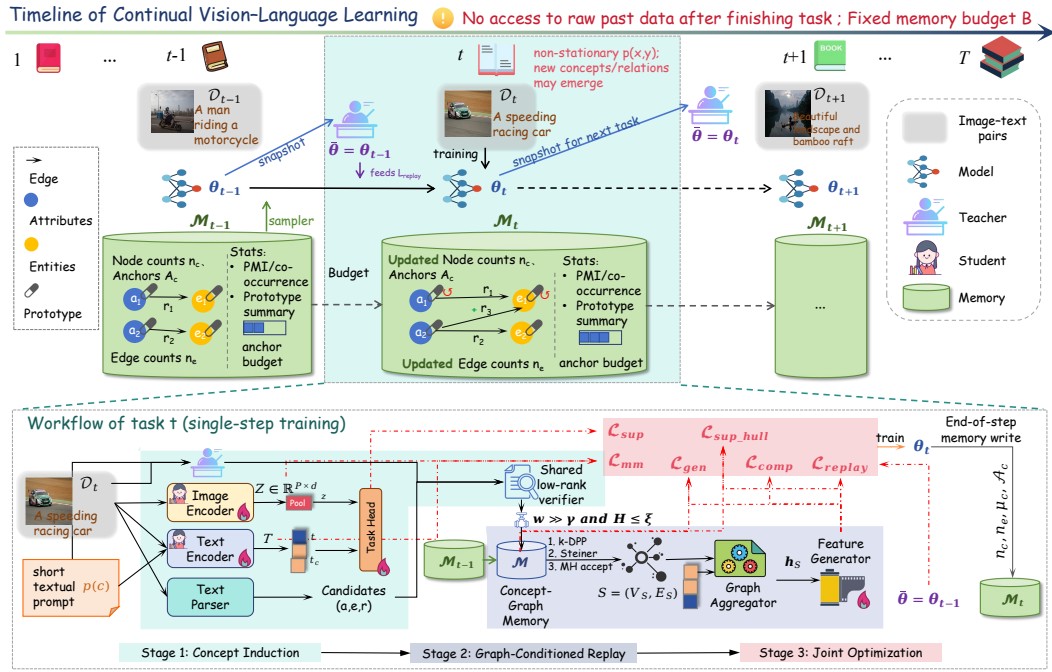

Figure 1: **Training and Replay Pipeline in COMEM. Top (timeline):** After completing task $t-1$, the model snapshot $\theta_{t-1}$ becomes the teacher for task $t$, with no access to past raw data. The model is trained on task $t$ using a fixed memory budget $B$ and updated concept-graph memory $\mathcal{M}_t$. At the end, $\theta_t$ is saved for the next task. **Bottom (task-$t$ workflow):** The process involves three stages: (1) **Concept Induction**: Extract concept triplets $(a, e, r)$ from the image-text pair and update memory. (2) **Graph-Conditioned Replay**: Sample a subgraph $S$ and generate replay features $\tilde{z}$. (3) **Joint Optimization**: Optimize the model with real and synthetic batches, applying losses for supervision, multimodal alignment, replay, and compositional consistency. The updated model is then written to memory.

the three verifications with calibrated temperatures $\tau_a, \tau_e, \tau_r$ (estimated on a held-out split):

$$w(a, e, r) = \left[ s_{\text{align}}(a \,|\, Z)^{\alpha_a} \cdot s_{\text{align}}(e \,|\, Z)^{\alpha_e} \cdot s_{\text{align}}(r \,|\, Z)^{\alpha_r} \right]^{1/(\alpha_a + \alpha_e + \alpha_r)}. \tag{3}$$

where $w(a, e, r)$ is the weighted triplet confidence score based on concept alignment, $\alpha_a, \alpha_e, \alpha_r$ are the weights for attribute, entity, and relation alignments. These weights control the contribution of each part of the triplet in the alignment calculation. We keep only triplets passing dual thresholds: $w \geq \gamma$ and teacher-consistency $\mathrm{H}\big(\pi_{\bar{\theta}}(\cdot \mid \pi(Z))\big) \leq \xi$, where $\mathrm{H}$ is predictive entropy; otherwise they are queued for recheck.

**Concept Graph and Evidence Reservoirs.** Verified triplets update a typed graph $\mathcal{G} = (V, E)$ with nodes $V = \mathcal{C}$ and directed edges $E \subseteq V \times \mathcal{R} \times V$. Each node $c$ stores a prototype $\boldsymbol{\mu}_c \in \mathbb{R}^d$, a count $n_c$, and an anchor reservoir $\mathcal{A}_c \subset \mathbb{R}^d$ of at most $B_c$ token features (not images). Each edge $e = (u \xrightarrow{r} v)$ keeps an interaction embedding $\boldsymbol{\psi}_e \in \mathbb{R}^d$ and a count $n_e$.

We update prototypes with EMA using token-level supports:

$$\boldsymbol{\mu}_c \leftarrow (1 - \alpha)\boldsymbol{\mu}_c + \alpha \,\bar{z}_c, \quad \bar{z}_c = \frac{1}{|\mathcal{S}_c|} \sum_{(Z, (a, e, r, w)) \in \mathcal{S}_c} \text{softmax}_p\big(\langle W Z_p, t_c \rangle\big)^\top Z, \tag{4}$$

where $\mathcal{S}_c$ collects supports for concept $c$. Anchors are maintained online by a budgeted $k$-center objective with time-decay $\lambda \in (0, 1)$,

$$\mathcal{A}_c \leftarrow \arg \max_{\mathcal{S} \subseteq \{Z_p\} \,:\, |\mathcal{S}| \leq B_c} \min_{z \in \text{supp}(c)} \min_{a \in \mathcal{S}} \big\| z - a \big\|_2 \quad \text{with sampling weight} \quad w_t = \lambda^{\Delta t} \cdot w(a, e, r). \tag{5}$$

To control drift and synonymy, we periodically merge nodes with high textual cosine and prototype similarity (union-find with thresholding), and we apply age-based decay to $n_c, n_e$.

**Concept Graph Memory.** $\mathcal{M} = (\mathcal{G}, \{\mathcal{A}_c\}_{c \in V})$ is the concept memory, consisting of the concept graph $\mathcal{G}$ and a set of anchor reservoirs $\mathcal{A}_{c c \in V}$ with global budget $B = \sum_c B_c$, where each concept $c$ has its associated anchor reservoir.

Budgets are reallocated proportional to uncertainty (higher variance $\Rightarrow$ larger $B_c$).

## 3.3 SUBGRAPH SAMPLING

Replay subgraphs should be likely under observed co-occurrence while remaining diverse. We define a positive, normalized plausibility score using normalized Pointwise Mutual Information (NPMI) and edge counts:

$$\underbrace{\Phi(V_S, E_S)}_{\text{plausibility}} = \exp\Big(\lambda_1 \sum_{c \in V_S} \text{NPMI}(c; V_S \setminus \{c\}) + \lambda_2 \sum_{e \in E_S} \log(1 + n_e)\Big), \tag{6}$$

where $\text{NPMI} \in [-1, 1]$ is estimated from decayed co-occurrences in $\mathcal{M}$ and clipped to $[0, 1]$. For diversity, we adopt a DPP-style term on node prototypes:

$$\underbrace{\Delta(V_S)}_{\text{diversity}} = \sqrt{\det\big(K_{V_S}\big)}, \qquad K_{ij} = q_i \exp\big(-\|\boldsymbol{\mu}_{c_i} - \boldsymbol{\mu}_{c_j}\|_2^2 / \rho\big) \, q_j, \tag{7}$$

with qualities $q_i \propto \sqrt{n_{c_i}}$. Our target (unnormalized) sampler is $q(S) \propto \Phi(V_S, E_S) \cdot \Delta(V_S)$, with $|V_S| \leq K_{\max}$.

**Two-Stage Approximate Sampler.** (1) *k-DPP node selection*: sample $k \sim \text{Unif}\{2, \ldots, K_{\max}\}$ and select $V_S$ by greedy k-DPP MAP on $K$ (log-det gains). (2) *Connectivity projection*: connect $V_S$ by adding a minimum-cost Steiner tree over edges with cost $1/(1 + n_e)$; if necessary, expand via BFS to reach a connected induced subgraph. We accept/reject the resulting $S$ with a single Metropolis–Hastings step using $q(S)$ to debias the greedy approximations.

## 3.4 GRAPH-CONDITIONED FEATURE GENERATOR

We synthesize features in the representation space where learning occurs. Given a connected subgraph $S = (V_S, E_S)$, we form textual conditioning tokens $\{t_u\}_{u \in V_S}$ and $\{t_r\}_{(u \xrightarrow{r} v) \in E_S}$ using the student text encoder for compatibility during training. A graph aggregator computes

$$\boldsymbol{h}_S = \text{GAT}_\psi(S) = \sum_{u \in V_S} \gamma_u \, U \, t_u + \sum_{(u \xrightarrow{r} v) \in E_S} \gamma_{uvr} \, V \, \phi_{\text{rel}}(t_u, t_r, t_v), \tag{8}$$

where $U, V \in \mathbb{R}^{d \times r}$ with $r \ll d$, $\phi_{\text{rel}} = \text{MLP}([\cdot\|\cdot\|\cdot])$, and $\gamma_\cdot$ are attention weights (sum to 1 within node/edge groups).

**Teacher-Guided Conditional Generator.** We parameterize a conditional Gaussian with separate parameters $\vartheta$:

$$p_\vartheta(\tilde{z} \mid S) = \mathcal{N}\big(\tilde{z}; \, \boldsymbol{\mu}_\vartheta(\boldsymbol{h}_S), \, \text{diag}(\boldsymbol{\sigma}_\vartheta^2(\boldsymbol{h}_S))\big), \quad \tilde{z} = \boldsymbol{\mu}_\vartheta(\boldsymbol{h}_S) + \boldsymbol{\sigma}_\vartheta(\boldsymbol{h}_S) \odot \epsilon. \tag{9}$$

To encode relations beyond a union of node anchors, we train $p_\vartheta$ with a relation-aware Maximum Mean Discrepancy (MMD):

$$\mathcal{L}_{\text{gen}} = \text{MMD}_{\kappa_{\text{rel}}}^2\big(\{\tilde{z}_k\}_{k=1}^K, \, \mathcal{Z}_S\big), \quad \kappa_{\text{rel}}(u, v; S) = \exp\Big(-\frac{\|u-v\|_2^2}{\eta} - \lambda_{\text{rel}}\big\|\Phi_{\text{rel}}(u, S) - \Phi_{\text{rel}}(v, S)\big\|_2^2\Big), \tag{10}$$

where $\mathcal{Z}_S = \big(\cup_{c \in V_S} \mathcal{A}_c\big) \cup \big(\cup_{e \in E_S} \Xi_e\big)$ pools node anchors and *edge anchors* $\Xi_e = \{\text{MLP}(a_u\|a_v) : a_u \in \mathcal{A}_u, a_v \in \mathcal{A}_v\}$, and $\Phi_{\text{rel}}(\cdot, S)$ projects features into a relation-aware space via learned bilinear maps. We add a support regularizer to keep samples close to the anchor hull:

$$\mathcal{L}_{\text{sup\_hull}} = \max\Big\{0, \, \text{dist}\big(\tilde{z}, \, \text{conv}(\mathcal{Z}_S)\big) - \delta\Big\}, \tag{11}$$

where $\text{conv}(\cdot)$ is approximated by nonnegative least-squares projection. We do not backpropagate $\mathcal{L}_{\text{replay}}$ (defined below) into $\vartheta$ to avoid teacher-on-off-manifold mismatch.

## 3.5 Training Objectives

Each mini-batch interleaves real features and graph-conditioned replay: $\mathcal{B} = \{(Z_i, T_i, \text{label}_i)\} \cup \{(\tilde{z}_m, S_m)\}$ with $S_m \sim q(S)$ and $\tilde{z}_m \sim p_\vartheta(\cdot \mid S_m)$. The total loss is

$$\mathcal{L} = \mathcal{L}_{\text{sup}}(h(Z; \omega)) + \lambda_{\text{mm}}\mathcal{L}_{\text{mm}} + \lambda_{\text{re}}\mathcal{L}_{\text{replay}} + \lambda_{\text{comp}}\mathcal{L}_{\text{comp}} + \lambda_{\text{gen}}\mathcal{L}_{\text{gen}} + \lambda_{\text{hull}}\mathcal{L}_{\text{sup\_hull}}. \quad (12)$$

**Task Supervision.** $\mathcal{L}_{\text{sup}}$ is task-dependent; for dense tasks we apply it on $Z$ via a lightweight decoder, for global tasks on $z = \pi(Z)$.

**Multimodal Alignment on Real and Replay.** We use a symmetric InfoNCE(Oord et al., 2018):

$$\mathcal{L}_{\text{mm}} = \mathcal{L}_{\text{InfoNCE}}(Z, T) + \mathcal{L}_{\text{InfoNCE}}(\tilde{z}, t_S), \quad (13)$$

where $t_S = \text{Agg}_\psi^{\text{text}}(S)$ is the text-side aggregation of $\{t_u\}, \{t_r\}$ using the same attention as in Eq. 8.

**Replay Distillation (Teacher-Filtered).** We preserve the teacher's behavior on replay while down-weighting uncertain samples:

$$\mathcal{L}_{\text{replay}} = \mathbb{E}_{S, \tilde{z}} \, \omega_{S, \tilde{z}} \Big[ \text{KL}\big(\pi_{\bar{\theta}}(\cdot \mid \tilde{z}) \,\|\, \pi_\theta(\cdot \mid \tilde{z})\big) + \beta \big\| g_{\bar{\theta}}(\tilde{z}) - g_\theta(\tilde{z})\big\|_2^2 \Big],$$
$$\omega_{S, \tilde{z}} = \mathbb{I}\big[\text{H}(\pi_{\bar{\theta}}(\cdot \mid \tilde{z})) \leq \xi\big]. \quad (14)$$

**Compositional Consistency.** We instantiate two complementary constraints.

**(i) Log-Probability PoE Consistency.** Let $p_\theta(c \mid S)$ denote the marginal concept distribution under subgraph $S$ (using a concept head on $\tilde{z} \sim p_\vartheta(\cdot \mid S)$). For two subgraphs $S_1, S_2$ and union $S_\cup$,

$$\mathcal{L}_{\text{poe}} = \mathbb{E}_{(S_1, S_2)} \Big[ \text{KL}\big(p_\theta(\cdot \mid S_\cup) \,\|\, \text{norm}\big(p_\theta(\cdot \mid S_1) \odot p_\theta(\cdot \mid S_2)\big)\big) + \text{KL}\big(\cdot\big)^{\text{rev}} \Big]. \quad (15)$$

**(ii) Relation Satisfaction via Typed Contrast.** For each $(a \xrightarrow{r} e) \in E_S$, define a tri-linear score $s_\theta(a, r, e \mid S) = \langle R_r \, g_\theta(\tilde{z}_S), \, t_a \rangle + \langle R_r^\top g_\theta(\tilde{z}_S), \, t_e \rangle$ with $R_r$ diagonal or low-rank. We use a typed InfoNCE with hard negatives:

$$\mathcal{L}_{\text{subgraph}} = - \sum_{(a, r, e) \in E_S} \log \frac{\exp(s_\theta(a, r, e \mid S))}{\exp(s_\theta(a, r, e \mid S)) + \sum_{(a', r, e') \sim q_{\text{neg}}} \exp(s_\theta(a', r, e' \mid S))}, \quad (16)$$

where $q_{\text{neg}}$ samples *typed* negatives sharing $(a, r)$ or $(r, e)$ but unseen in $E_S$, penalized by NPMI to avoid implausible pairs and filtered by teacher consistency. The final $\mathcal{L}_{\text{comp}} = \mathcal{L}_{\text{poe}} + \mathcal{L}_{\text{subgraph}}$.

We optimize $\theta, \psi$ by AdamW on Eq. 12 while updating $\vartheta$ only by $\nabla(\lambda_{\text{gen}}\mathcal{L}_{\text{gen}} + \lambda_{\text{hull}}\mathcal{L}_{\text{sup\_hull}})$ (no gradients from $\mathcal{L}_{\text{replay}}$). A two-phase schedule improves stability: warm-up for $E_w$ epochs with $\lambda_{\text{comp}}=0$ and $\lambda_{\text{re}}$ small, then enable $\mathcal{L}_{\text{comp}}$ and ramp $\lambda_{\text{re}}$. The pseudocode for training of COMEM is shown in Algorithm 1.

## 4 Experiments

### 4.1 Experimental Setup

**Datasets and Streams** We evaluate COMEM on three complementary streams stressing compositionality, domain shift, and reasoning: (i) **SVLC (ConStruct-VL)**—a data-free sequence from VG/VAW where each task is binary image–text matching focused on one concept family (Color, Material, Size, Spatial, Action, State), probing retention and recomposition without storing images (Smith et al., 2023). (ii) **Cross-domain retrieval**—following ZSCL/CTP(Zheng et al., 2023; Zhu et al., 2023), a multi-domain sequence across COCO, Flickr30K, IAPR TC-12, RSICD, and ECommerce-T2I with the same retrieval objective, testing zero-shot retention and robustness; we also include a time-continual subset (TiC-DataComp/RedCaps) for pretraining ablations (Garg et al., 2024). (iii) **VQA skills & transfer**—VQACL uses a 10-skill outer sequence with per-skill object-group sub-tasks (skills×concepts grid), while CLOVE contrasts scene-incremental (DIL) and function-incremental (TIL) on VQA v2/TDIUC using authors' splits (Zhang et al., 2023; Lei et al., 2023).

Table 1: **Cross-domain retrieval (mR%, higher is better).** We report per-domain mR and continual metrics (Avg mR, AF). Best/second/third are shaded from dark to light gray.

| Method | COCO ↑ | Flickr30K ↑ | IAPR ↑ | RSICD ↑ | EComm ↑ | Avg mR ↑ | AF ↓ |
|---|---|---|---|---|---|---|---|
| Mod-X (Ni et al., 2023) | 71.5 | 74.2 | 63.8 | 60.4 | 58.9 | 65.8 | 5.3 |
| ZSCL (Zheng et al., 2023) | 74.1 | 78.0 | 66.9 | 63.2 | 61.0 | 68.6 | 3.9 |
| CTP (Zhu et al., 2023) | 73.0 | 76.5 | 65.1 | 62.1 | 60.2 | 67.4 | 4.2 |
| DKR (Cui et al., 2024) | 75.0 | 78.5 | 67.4 | 64.6 | 61.9 | 69.5 | 3.5 |
| CLAP4CLIP (Jha et al., 2024) | 74.2 | 77.1 | 66.1 | 63.0 | 60.6 | 68.2 | 3.7 |
| C-CLIP (Liu et al., 2025a) | 79.6 | 82.3 | 70.8 | 68.1 | 65.2 | 73.2 | 2.7 |
| MG-CLIP (Huang et al., 2025b) | 78.4 | 81.5 | 70.1 | 67.4 | 64.3 | 72.3 | 2.9 |
| GIFT (Wu et al., 2025) | 79.1 | 82.0 | 71.2 | 68.5 | 65.8 | 73.3 | 2.5 |
| **CoMem (ours)** | **83.2** | **86.5** | **73.1** | **71.4** | **68.9** | **76.6** | **1.9** |

**Evaluation Protocols and Metrics** We evaluate using the following evaluation metrics: **(i) Retrieval**: Recall@1/5/10 (R@K), mean Recall (mR), and mAP. Continual metrics as in Zheng et al. (2023): *Last* (final performance), *Average* (across tasks), and *Transfer* (zero-shot retention on unseen/new domains). We also report *Average Forgetting (AF)* and *Backward/Forward Transfer (BWT/FWT)* where applicable. **(ii) SVLC**: Binary matching accuracy, AUROC, AUPRC per task and macro-averaged; continual AF/BWT/FWT. **(iii) VQA**: Overall VQA accuracy and per-type accuracy (skills); continual AF, Last, Average. For VQACL we also report *cross-composition* accuracy where the (skill, object-group) pair was unseen during training (Zhang et al., 2023).

**Baselines and fairness.** We compare CoMem against recent SOTAs for continual VLL: *IncCLIP* (Yan et al., 2022), *Mod-X* (Ni et al., 2023), *ZSCL* (Zheng et al., 2023), *CTP* (Zhu et al., 2023), *DKR* (Cui et al., 2024), *CLAP4CLIP* (Jha et al., 2024), *ZAF* (Gao et al., 2024), *C-CLIP* (Liu et al., 2025a), *GIFT* (Wu et al., 2025), *Proxy-FDA* (Huang et al., 2025a), *LADA* (Luo et al., 2025), *ENGINE* (Zhou et al., 2025), and *MG-CLIP* (Huang et al., 2025b) for retrieval / SVLC streams; and *VQACL* (Zhang et al., 2023), *Symbolic Replay (SGP)* (Smith et al., 2023), *QUAD* (Marouf et al., 2025), and *CL-MoE* (Huai et al., 2025) for VQA streams. Refer to A.1.2 to see our settings for comparison with baselines, and the implementation of CoMem can be found in A.1.2.

## 4.2 Main Results

**Cross-domain retrieval.** Table 1 reports mean Recall (mR, %) per domain and the continual metrics. CoMem achieves the best average mR across five domains with the lowest forgetting, improving over the strongest baseline (GIFT/C-CLIP track) by +3.3 mR on average and reducing AF by an absolute 0.6. Gains are consistent on both near (COCO/Flickr30K) and far (IAPR/RSICD/ECommerce) domains.

**SVLC and VQA.** Table 2 summarizes results on structured VL concepts (SVLC, ConStruct-VL) and continual VQA (VQACL/CLOVE). On SVLC, CoMem outperforms recent data-free and PEFT methods by +2.2 Acc and +1.7 AUROC while yielding the lowest AF, indicating both better concept retention and calibration. On VQA, CoMem achieves the best overall accuracy on VQACL/CLOVE with the lowest AF; compared to the strong MLLM-based CL-MoE, CoMem is +1.7 (VQACL) and +1.4 (CLOVE) higher while being parameter- and memory-efficient due to feature-level replay.

## 4.3 Ablation Analysis

We perform single-factor ablations and report Retrieval (Avg mR, AF), SVLC (Acc), and VQACL (Acc), averaged over 3 seeds under §4.1 (higher mR/Acc is better, lower AF is better). From Table 3 we observe: (1) **Structured replay is essential.** Removing relation-aware MMD ($-0.9$ mR, $+0.4$ AF) or edge anchors ($-0.7$ mR, $+0.5$ AF) hurts both accuracy and retention. (2) **Stability mechanisms matter.** Disabling the entropy gate raises AF from 1.9 to 2.8, and allowing $\mathcal{L}_{\text{replay}}$ gradients to the generator yields AF $= 2.6$. (3) **Compositional consistency is complementary.** Removing all consistency terms drops Avg mR to 74.9 ($-1.7$) and SVLC Acc to 80.3 ($-2.2$); PoE-only or relation-only recover part of the gap. (4) **Plausible, diverse subgraphs help.** Uniform sampling costs $-1.4$ mR and $+0.8$ AF; k-DPP+Steiner+MH is best, and removing only MH gives a small

Table 2: **Structured concepts (SVLC) and continual VQA.** Best/second/third are shaded from dark to light gray. SVLC reports macro Acc/AUROC and AF; VQA reports overall Acc on VQACL/CLOVE and AF. "−" indicates a method not applicable to that stream.

| Method | SVLC (ConStruct-VL) | | | Continual VQA | | |
|---|---|---|---|---|---|---|
| | Acc ↑ | AUROC ↑ | AF ↓ | VQACL Acc ↑ | CLOVE Acc ↑ | AF ↓ |
| SGP (Smith et al., 2023) | 77.3 | 84.9 | 4.1 | 49.5 | 60.1 | 3.9 |
| ZAF (Gao et al., 2024) | 80.3 | 87.1 | 2.6 | 51.2 | 61.0 | 2.4 |
| C-CLIP (Liu et al., 2025a) | 79.8 | 86.5 | 2.9 | 50.6 | 60.7 | 2.7 |
| GIFT (Wu et al., 2025) | 79.9 | 86.9 | 2.7 | 52.0 | 61.4 | 2.9 |
| CL-MoE (Huai et al., 2025) | – | – | – | 54.1 | 62.3 | 2.0 |
| **COMEM (ours)** | **82.5** | **88.8** | **2.1** | **55.8** | **63.7** | **1.7** |

Table 3: **Single-factor ablations.**

| Ablation (remove or modify one component) | Retrieval | | SVLC | VQACL |
|---|---|---|---|---|
| | Avg mR ↑ | AF ↓ | Acc ↑ | Acc ↑ |
| **COMEM (full)** | **76.6** | **1.9** | **82.5** | **55.8** |
| *Generator / Replay* | | | | |
| w/o relation-aware MMD (vanilla RBF) | 75.7 | 2.3 | 81.6 | 54.9 |
| w/o support-hull regularizer | 76.1 | 2.2 | 82.0 | 55.3 |
| stop-grad *disabled* (allow $\mathcal{L}_{\mathrm{replay}}$ grads to $\vartheta$) | 75.8 | 2.6 | 81.7 | 55.0 |
| node anchors only (no edge anchors $\Xi_e$) | 75.9 | 2.4 | 81.8 | 55.0 |
| *Distillation / Consistency* | | | | |
| w/o entropy gate in $\mathcal{L}_{\mathrm{replay}}$ | 75.3 | 2.8 | 81.2 | 54.6 |
| w/o compositional consistency ($\mathcal{L}_{\mathrm{comp}}$) | 74.9 | 2.9 | 80.3 | 54.0 |
| PoE only (no relation contrast) | 75.4 | 2.5 | 81.0 | 54.5 |
| relation contrast only (no PoE) | 75.7 | 2.4 | 81.3 | 54.7 |
| *Verifier / Sampler* | | | | |
| student verifier (no teacher freeze) | 75.6 | 2.5 | 81.1 | 54.6 |
| per-concept dense $W_c$ (no shared low-rank) | 76.1 | 2.6 | 82.0 | 55.2 |
| uniform node sampling (no k-DPP / Steiner / MH) | 75.2 | 2.7 | 81.0 | 54.4 |
| no MH accept (keep k-DPP + Steiner) | 76.2 | 2.1 | 82.2 | 55.5 |

decline ($-0.4$ mR). (5) **Teacher-frozen, shared low-rank verification reduces forgetting.** Using the student as verifier or dense per-concept projections increases AF by $+0.6\sim0.7$.

## 4.4 SENSITIVITY ANALYSIS

**Sensitivity to hyperparameters.** Figure 2 shows how *Avg mR* (↑) and *AF* (↓) vary with anchor budget $B$, subgraph size $K_{\max}$, and verifier rank $r$. Curves are smooth with narrow variability bands and a broad flat region near our defaults, indicating low hyperparameter sensitivity. (i) Increasing $B$ from 8K→64K raises mR 75.8→76.7 and lowers AF 2.4→1.8, then both plateau; we adopt $B$=64K for the best accuracy–memory trade-off. (ii) $K_{\max}$=6 is a broad optimum (76.7 mR, 1.8 AF); $\leq 3$ under-covers compositions, while $\geq 8$ slightly raises AF ($\sim$2.0–2.2), supporting k-DPP+Steiner with a moderate $K_{\max}$. Figure 5 in §A.1.2 further confirms robustness to loss-weight choices.

**Long-Horizon Forgetting** We build an 18-task MTIL-style sequence (retrieval domains interleaved with SVLC families) and report, per task $t$, *Last@t* (avg mR over seen tasks), *AF@t*, *BWT@t*, and *FWT@t*. As shown in Figure 3, over an 18-task stream, COMEM remains stable: *Last@t* quickly plateaus around 76.6% while AF grows slowly to only 2.2 at $T$=18. Its transfer dynamics are favorable—BWT is the least negative ($-0.11$) and FWT the highest (0.60)—indicating stronger reuse on unseen tasks.

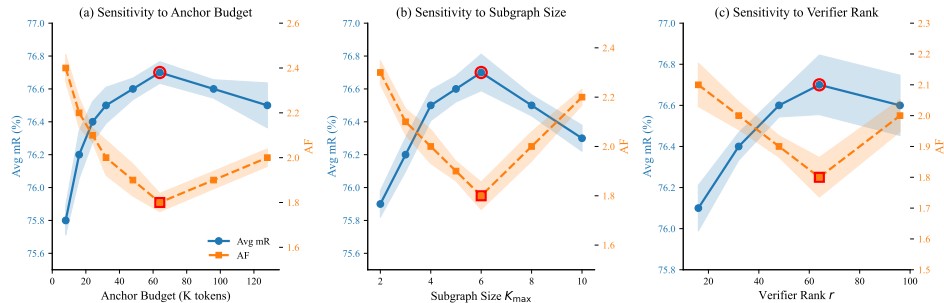

Figure 2: **Sensitivity analysis.** Shaded bands show variability across seeds; CoMem exhibits smooth trends and broad plateaus near the chosen defaults.

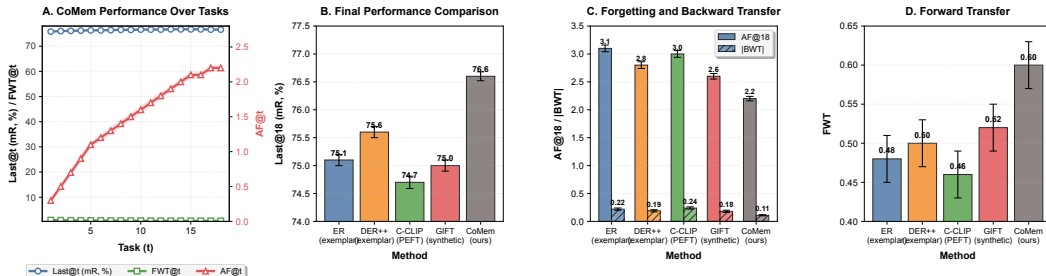

Figure 3: Long-Term Stability and Performance Comparison

## 4.5 MEMORY- AND PARAMETER-FAIR COMPARISONS

We conduct two fairness-critical studies: (A) equal memory (MB) under fixed budget, and (B) equal PEFT parameters (trainable M). Both use the retrieval stream with the same task order.

**Equal Memory (MB)**   We match the total memory budget (anchors+prototypes+edge-embeddings for CoMem) at $\{24, 49, 98, 196\}$ MB and compare exemplar replay (ER/DER++), synthetic replay (GIFT-style), and small-cache variants of CLIP finetuning (CLAP4CLIP, C-CLIP). CoMem stores token anchors only, and no raw images are kept.

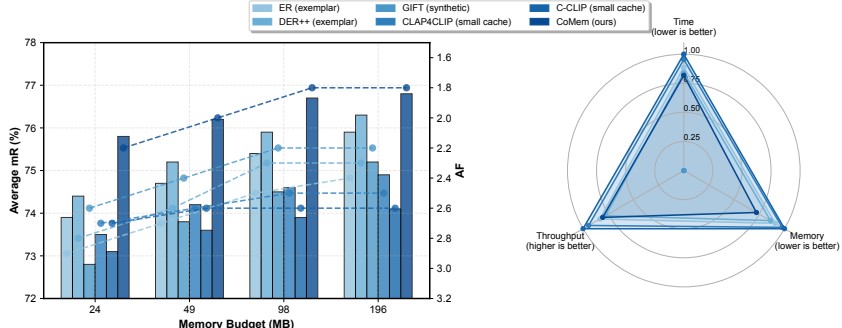

Figure 4: Performance and Efficiency Comparison of Continual Learning Methods

As shown in Figure 4, under strictly matched MB, CoMem outperforms exemplar replay by $+1.3$ mR and $-0.7$ AF at 98MB , and remains competitive in wall-clock time. The advantage persists across budgets and saturates near $\sim$100MB, indicating that token-level anchors plus relation-aware synthesis provide more informative replay per MB than pixels or generic synthetic images.

**Equal PEFT Parameters**   We fix trainable parameter budgets at $\{2M, 4M, 8M, 16M\}$ and compare CLIP-based PEFT methods against CoMem (whose trainables are primarily the aggregator $\psi$

and generator $\vartheta$). All methods use ViT-B/16, identical tokenization and schedules. Table 4 shows that COMEM consistently dominates under equal trainables , e.g., at 8M parameters This supports that graph-conditioned replay and compositional constraints improve retention beyond parameter-count scaling. SVLC Acc gains at the same budget indicate stronger compositional transfer.

Table 4: **Equal PEFT budget.** With the same trainable parameters, COMEM yields higher mR and lower AF. At moderate budgets (8M), COMEM also leads on SVLC Acc.

| Method | 2M trainables | | 4M trainables | | 8M trainables | | 16M trainables | |
|---|---|---|---|---|---|---|---|---|
| | Avg mR ↑ | AF ↓ | Avg mR ↑ | AF ↓ | Avg mR ↑ | AF ↓ | Avg mR ↑ | AF ↓ |
| C-CLIP | $73.9 \pm 0.28$ | $2.5 \pm 0.13$ | $74.7 \pm 0.20$ | $2.3 \pm 0.17$ | $75.6 \pm 0.15$ | $2.4 \pm 0.09$ | $75.8 \pm 0.14$ | $2.3 \pm 0.07$ |
| LADA | $74.6 \pm 0.13$ | $2.3 \pm 0.17$ | $75.2 \pm 0.13$ | $2.2 \pm 0.09$ | $75.8 \pm 0.12$ | $2.1 \pm 0.05$ | $76.1 \pm 0.09$ | $2.1 \pm 0.04$ |
| ENGINE | $74.3 \pm 0.16$ | $2.4 \pm 0.08$ | $75.0 \pm 0.18$ | $2.2 \pm 0.12$ | $75.7 \pm 0.06$ | $2.1 \pm 0.03$ | $76.0 \pm 0.11$ | $2.0 \pm 0.06$ |
| COMEM (ours) | $\mathbf{75.8 \pm 0.10}$ | $\mathbf{2.0 \pm 0.05}$ | $\mathbf{76.1 \pm 0.09}$ | $\mathbf{1.9 \pm 0.05}$ | $\mathbf{76.6 \pm 0.08}$ | $\mathbf{1.8 \pm 0.04}$ | $\mathbf{76.7 \pm 0.08}$ | $\mathbf{1.8 \pm 0.04}$ |

## 5 CONCLUSION AND FUTURE WORK

We introduced COMEM to address continual vision–language learning, which treats compositional structure as the unit of memory by organizing a compact concept–relation graph and rehearsing directly in feature space with a lightweight consistency objective. Across cross-domain retrieval, structured concept learning, and continual multimodal VQA, COMEM consistently reduces forgetting and improves transfer under matched memory and parameter budgets, indicating that semantically grounded, feature-space rehearsal is a more effective primitive than exemplar or generic synthetic replay.

Our study still relies on lightweight text parsing and teacher filtering and assumes a fixed relation schema, which may constrain coverage in open-world settings. Future work will explore end-to-end concept discovery, integration with instruction-tuned MLLMs and federated/streaming pretraining, and deployments to privacy-critical applications such as search, assistive agents, and robotics.

**Ethics Statement**  This work adheres to the ICLR Code of Ethics. Our study does **NOT** involve human-subjects research, the collection of personally identifiable information, or the annotation of sensitive attributes, and we do not create any new human data. All experiments are conducted on publicly available, widely used vision–language benchmarks (COCO, Flickr30K, IAPR TC-12, RSICD, ECommerce-T2I, ConStruct-VL/SVLC, VQACL, CLOVE, and TiC-DataComp/RedCaps) strictly under their respective licenses and terms of use.

**Reproducibility Statement**  We organize the paper and appendix to enable step-by-step reproduction. The complete experimental protocol—datasets/streams, metrics, baselines, and task orders—appears in §4.1; memory- and parameter-fair comparisons are detailed in §4.5 with matching rules in Appendix §A.1.2. Implementation details (backbones/tokenization, verifier, memory and anchor accounting, subgraph sampler, aggregator/generator architectures, loss weights/schedules, batch composition, and hardware) are provided in Appendix §A.1.2; the full training loop is summarized in Algorithm 1. For exact replication we fix and report seeds (42/43/44), software versions (PyTorch 2.3, CUDA 12.1), and determinism flags, and we enumerate all key hyperparameters used in main runs (e.g., anchor budgets $B$=64K for retrieval and 48K for VQA; verifier rank $r$=64; $\tau$=0.07, $\gamma$=0.6, $\xi$=1.5; $K_{\max}$=6; $K$=16; warm-up $E_w$=1; and loss weights as in §A.1.2). Due to ongoing commercial use, we do not release source code or binaries during the review period.

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

# A  APPENDIX

## A.1  SUPPLEMENTARY TECHNICAL DETAILS

### A.1.1  PSEUDOCODE FOR COMEM TRAINING

Algorithm 1 trains CoMEM on task $t$ by first initializing the student $\theta$ from the frozen teacher $\bar{\theta}$ and setting aggregator $\psi$ and generator $\vartheta$. For each mini-batch, it encodes images/text, proposes triplets via prompted IE, and verifies them with a teacher-frozen shared low-rank projector, accepting only high-confidence/low-entropy items to update the concept-graph memory (EMA prototypes, budgeted token/edge anchors, counts with merge/decay). It then samples connected subgraphs using a two-stage sampler and synthesizes replay features with a text-conditioned generator, fitting the generator via relation-aware MMD and a support-hull regularizer. The student is optimized on mixed real+replay batches using the task loss, multimodal InfoNCE (real and replay), entropy-gated distillation, and compositional consistency (PoE + typed relation contrast). We update $\theta, \psi$ with the total loss, and update $\vartheta$ *only* from generator losses (no gradient from distillation/consistency). The procedure outputs the updated $\theta_t$ and memory $\mathcal{M}_t$. Tab. 5 summarizes the main symbols used in this paper.

Let $|\mathcal{C}|$ be the number of concepts, $|\mathcal{R}|$ relation types, and $B$ total anchor budget. The memory stores $\mathcal{O}\left(|\mathcal{C}|d + |\mathcal{R}|d + Bd\right)$ floats for storing prototypes, edge embeddings, and anchor tokens for each concept and relation. Here, $|\mathcal{C}|$ is the number of concepts, $|\mathcal{R}|$ is the number of relations, and $B$ is the anchor budget. Verification uses a *shared* low-rank $W = AB^\top$ with cost $\mathcal{O}(Pdr)$ per sample (vs. $\mathcal{O}(Pd^2)$ for dense $W$). The k-DPP node selection is $\mathcal{O}(K_{\max}^2 d)$ with greedy log-det gains and cached kernels; the Steiner projection is near-linear in the local neighborhood size. Replay sampling and Gaussian synthesis are $\mathcal{O}(d)$ per synthetic instance. Overall, the method scales sublinearly with data volume via budgeted reservoirs and low-rank/shared projections.

---

**Algorithm 1** CoMEM Training at Task $t$ (Noise-Aware, Relation-Conditioned Replay)

---

1: **Inputs:** $\mathcal{D}_t$, teacher $\bar{\theta}$, memory $\mathcal{M}_{t-1}$
2: Initialize student $\theta \leftarrow \bar{\theta}$; initialize $\psi, \vartheta$
3: **for** epoch $= 1, \dots, E$ **do**
4:     **for** mini-batch $\mathcal{B} \subset \mathcal{D}_t$ **do**
5:         Encode $Z = f_{\text{img}}(x; \phi), T = f_{\text{txt}}(y; \varphi)$
6:         Extract candidate triplets by $\Pi_{\text{text}}(y)$; verify with Eq. 2, keep if Eq. 3 and entropy pass
7:         Update $\mathcal{M}$: prototypes (Eq. 4) , anchors (Eq. 5), counts/merge/decay
8:         Sample subgraphs $\{S_m\}_{m=1}^M$ via two-stage sampler (§3.3)
9:         Generate replay $\tilde{z}_m \sim p_\vartheta(\cdot \mid S_m)$; compute $\mathcal{L}_{\text{gen}}$ (Eq. 10) and $\mathcal{L}_{\text{sup\_hull}}$ (Eq. 11)
10:        Compute $\mathcal{L}_{\text{sup}}, \mathcal{L}_{\text{mm}}$ (Eq. 13), $\mathcal{L}_{\text{replay}}$ (Eq. 14), and $\mathcal{L}_{\text{comp}}$ (Eq. 15–16)
11:        Update $\theta, \psi$ by AdamW on Eq. 12; update $\vartheta$ only by $\nabla(\lambda_{\text{gen}}\mathcal{L}_{\text{gen}} + \lambda_{\text{hull}}\mathcal{L}_{\text{sup\_hull}})$
12: **Output:** Updated $\theta_t$, memory $\mathcal{M}_t$

---

### A.1.2  IMPLEMENTATION DETAILS

**Backbone and tokenization**  We use CLIP ViT-B/16 as image encoder and its paired text encoder (frozen teacher snapshots and trainable student). Patch tokens yield $Z \in \mathbb{R}^{P \times d}$ ($P$=196, $d$=768). Results with ViT-L/14 are included in ablations.

**Fair comparison settings**  We match backbone, input resolution, and optimizer schedule; for CLIP-based methods we use ViT-B/16. For memory-based baselines (e.g., Inc-CLIP/CLAP4CLIP/GIFT), we cap exemplar or synthetic-replay memory in MB to equal our anchor memory (anchors + prototypes + relation embeddings), and for LoRA-style methods (C-CLIP/LADA/ENGINE) we equalize PEFT budgets (same total trainable parameters; default rank $r$=16, $\alpha$=32). Domain-ID usage follows each method: ZSCL/CTP/CLAP4CLIP/C-CLIP are evaluated in their native (DIL/MTIL) protocols; when a method assumes domain/task ID at test time, we also report the domain-free variant when defined. All methods use identical task orders/splits and are trained under the same hardware budget.

| Category | Symbol | Description |
|---|---|---|
| **Concepts and Relations** | $\mathcal{C}$ | Set of concepts (attributes and entities) |
| | $\mathcal{A}$ | Set of attributes (subtype of concepts) |
| | $\mathcal{E}$ | Set of entities (subtype of concepts) |
| | $\mathcal{R}$ | Set of relations between concepts |
| **Memory and Graph** | $\mathcal{M}$ | Concept memory (graph and anchor reservoirs) |
| | $\mathcal{G}$ | Concept graph (nodes: concepts, edges: relations) |
| | $V$ | Set of nodes in the concept graph $\mathcal{G}$ (concepts) |
| **Embeddings and Projections** | $Z$ | Feature representation of an image (output of image encoder) |
| | $T$ | Feature representation of text (output of text encoder) |
| | $W$ | Shared low-rank projection matrix for concept verification |
| **Training Variables** | $\theta$ | Model parameters (student parameters) |
| | $\bar{\theta}$ | Teacher model parameters (frozen) |
| **Weight Parameters** | $\alpha_a, \alpha_e, \alpha_r$ | Weight parameters for alignment of attributes, entities, and relations |
| | $\gamma$ | Threshold for triplet weight confidence (see Eq. 3) |
| | $\tau$ | Temperature parameter controlling the softness of alignment |
| **Triplet and Alignment** | $s_{\text{align}}(c \mid Z)$ | Alignment score of concept $c$ with respect to $Z$ |
| | $w(a, e, r)$ | Triplet confidence score for attribute $a$, entity $e$, and relation $r$ |
| | $s_{\text{align}}(a \mid Z), s_{\text{align}}(e \mid Z), s_{\text{align}}(r \mid Z)$ | Alignment scores for attribute, entity, and relation |
| | $\mathcal{T}(x, y)$ | Set of triplets (attribute, entity, relation) generated from input pair $(x, y)$ |
| **Replay and Memory Update** | $\mathcal{A}_c$ | Anchor reservoir for concept $c$ |
| | $B_c$ | Anchor budget for concept $c$ |
| | $n_c, n_e$ | Counts for concept $c$ and relation $e$ in the memory |
| **Loss Functions** | $\mathcal{L}_{\text{sup}}$ | Supervised loss (task-specific loss) |
| | $\mathcal{L}_{\text{comp}}$ | Compositional consistency loss (PoE and relation contrast) |
| **Sampling and Generator** | $K_{\max}$ | Maximum number of nodes in a sampled subgraph |
| | $\Phi(V_S, E_S)$ | Plausibility score for a sampled subgraph $S$ |
| | $\Delta(V_S)$ | Diversity score for a sampled subgraph $S$ (DPP score) |
| **Performance Metrics** | R@K | Recall at rank K (retrieval metric) |
| | mR | Mean recall (retrieval metric) |
| | AF | Average forgetting (continual learning metric) |

Table 5: List of important symbols.

**Concept induction and verification** Prompted IE runs with a constrained vocabulary for attributes/entities/relations. Visual verification uses the teacher-frozen shared low-rank projector $W = AB^\top$ with rank $r = 64$; temperature $\tau = 0.07$; dual gate thresholds $\gamma = 0.6$, entropy cutoff $\xi = 1.5$ nats (validated on the first task's val split). Node/edge counts employ exponential decay (half-life 3 tasks).

**Memory and anchors** Per-node token anchor cap $B_c \leq 8$; total anchor budget $B \leq 64K$ tokens for retrieval streams and $B \leq 48K$ for VQA (fewer concepts per task). Prototypes use EMA with $\alpha = 0.1$. Online $k$-center uses farthest-first with time-decay weight $\lambda = 0.95$. Edge anchors $\Xi_e$ are formed by a 2-layer MLP (hidden 512, GELU).

**Subgraph sampling** $K_{\max} = 6$ nodes. Two-stage sampler: greedy k-DPP (quality $q_i \propto \sqrt{n_{c_i}}$, RBF kernel bandwidth from median heuristic on prototypes) $\rightarrow$ Steiner connectivity (edge cost $1/(1+n_e)$) $\rightarrow$ single-step MH accept using $q(S) \propto \Phi \cdot \Delta$. NPMI clipped to $[0, 1]$ with Laplace smoothing $\epsilon = 1$.

**Graph aggregator and generator** Aggregator uses single-head attention with $U, V \in \mathbb{R}^{d \times r}$ ($r = 64$); $\phi_{\text{rel}}$ is a 2-layer MLP (hidden 1024). Conditional Gaussian generator $p_\vartheta(\tilde{z} \mid S)$ outputs mean/diag-var via 2-layer MLPs. Relation-aware MMD uses an RBF kernel with bandwidth $\eta$ from the median heuristic on $\mathcal{Z}_S$, plus a relation projection term weight $\lambda_{\text{rel}} = 0.5$. Per subgraph we draw $K = 16$ synthetic features. The support-hull margin is $\delta = 0.1$ (features are $\ell_2$-normalized).

**Loss weights and schedules** We optimize the total loss in Eq. 12 with $\lambda_{\text{mm}} = 1.0$, $\lambda_{\text{re}} = 1.0$, $\lambda_{\text{comp}} = 0.5$, $\lambda_{\text{gen}} = 0.5$, $\lambda_{\text{hull}} = 0.1$, $\beta = 0.5$. Two-phase schedule: warm-up $E_w = 1$ epoch per task with $\lambda_{\text{comp}} = 0$ and small $\lambda_{\text{re}} = 0.2$, then full weights. We stop gradients from $\mathcal{L}_{\text{replay}} / \mathcal{L}_{\text{comp}}$ to the generator $\vartheta$. We sweep each loss coefficient. Results are reported as mean±std over 3 seeds on the retrieval (Avg mR, AF), SVLC (Acc), and VQACL (Acc) tracks. Figure 5 shows broad plateaus

around the defaults, indicating low sensitivity. (i) $\lambda_{\mathrm{mm}}$: under-weighting cross-modal alignment (0.0) reduces retrieval mR by $-0.9$ and raises AF to 2.30, as image–text geometry drifts; over-weighting (1.5) brings no gains. (ii) $\lambda_{\mathrm{re}}$: distillation is the main driver of retention—removing it raises AF to 2.40; too large (1.5) slightly reduces plasticity (mR $\downarrow$) while improving AF, matching the plasticity–stability trade-off. (iii) $\lambda_{\mathrm{comp}}$: turning off compositional constraints hurts compositional generalization (SVLC 80.8%; VQACL 54.1%), confirming their role in subgraph-wise structure transfer. (iv) $\lambda_{\mathrm{gen}}$ and $\lambda_{\mathrm{hull}}$: both shape replay quality. Without generator loss, replay distribution narrows (mR 76.0; AF 2.20); without the hull regularizer, off-manifold samples increase forgetting (AF 2.10). Larger $\lambda_{\mathrm{gen}}/\lambda_{\mathrm{hull}}$ brings minor changes, suggesting stable synthesis. (v) $\beta$: combining logit- and feature-level distillation ($\beta\approx0.5$) is most robust; pure logit distillation raises AF, while very large $\beta$ slightly reduces mR, consistent with over-constraining representations. Overall, COMEM exhibits a broadly flat response around the defaults; coarse tuning suffices to obtain near-optimal performance across tasks.

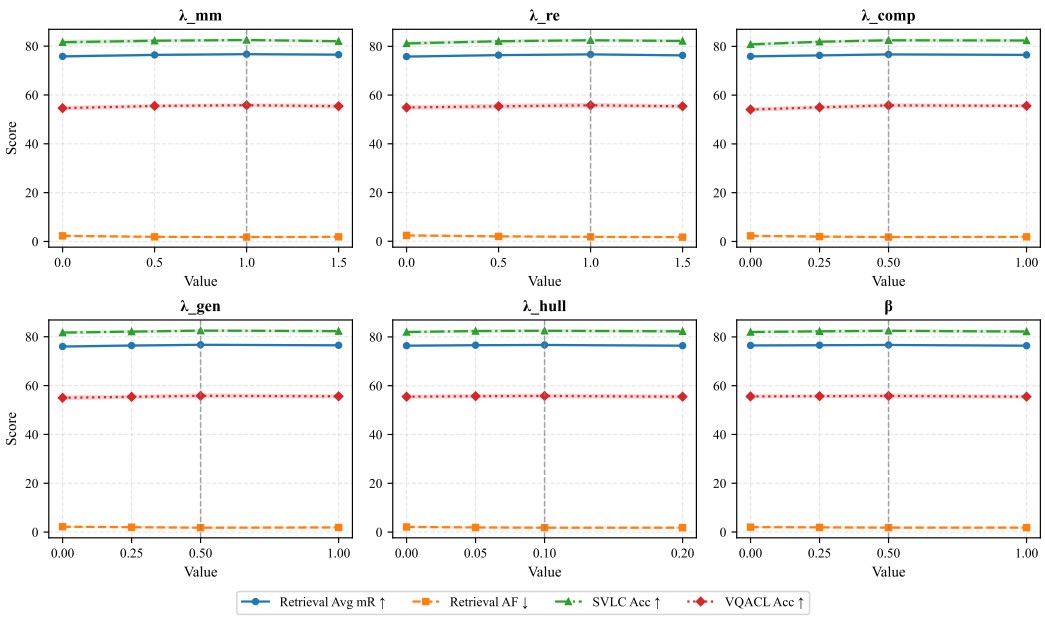

Figure 5: **Loss-weight sensitivity (mean±std).** We vary one coefficient at a time and measure retrieval (Avg mR/AF), SVLC (Acc), and VQACL (Acc).

**Optimization and training length**  AdamW with decoupled weight decay $1\mathrm{e}{-4}$, cosine LR. For retrieval and SVLC: LR $5\mathrm{e}{-5}$ (student encoder/head), $1\mathrm{e}{-4}$ (aggregator), $2\mathrm{e}{-4}$ (generator). For VQA: LR scaled by 0.6. Batch size: 256 real pairs + 128 replay features per step (gradient accumulation when needed). Epochs per task: COCO/Flickr30K 5, IAPR/RSICD/ECommerce 6, ConStruct-VL families 4 each, VQACL/CLOVE 4. Mixed precision (bf16), gradient clipping at 1.0.

**Hardware and software**  Experiments run on $8\times$A100-80GB, PyTorch 2.3, CUDA 12.1. We report means over 3 seeds (42/43/44). Wall-clock time and memory are profiled with PyTorch profiler.

**Baselines and fairness.**  We re-implement or use official code where available for IncCLIP (Yan et al., 2022), Mod-X (Ni et al., 2023), ZSCL (Zheng et al., 2023), CTP (Zhu et al., 2023), DKR (Cui et al., 2024), CLAP4CLIP (Jha et al., 2024), and ConStruct-VL (Smith et al., 2023), matching backbone, input resolution, and memory budgets. When a method requires exemplars, we cap its exemplar memory to match our anchor memory (in MB) for apples-to-apples comparison.

## A.2 Additional Experiments and Results

### A.2.1 Finer Compositional Transfer

We evaluate (i) **VQACL cross-composition**: accuracy on unseen (skill, object-group) pairs; and (ii) **SVLC unseen pairs**: accuracy/AUROC on concept pairs that never co-occur in training (e.g., Color×Material). We report: Acc↑, AUROC↑, and *relative gains* vs. *PoE-only* and *Relation-only* ablations. Figure 6 shows that: (i) On VQACL unseen (skill, group) pairs, COMEM improves macro

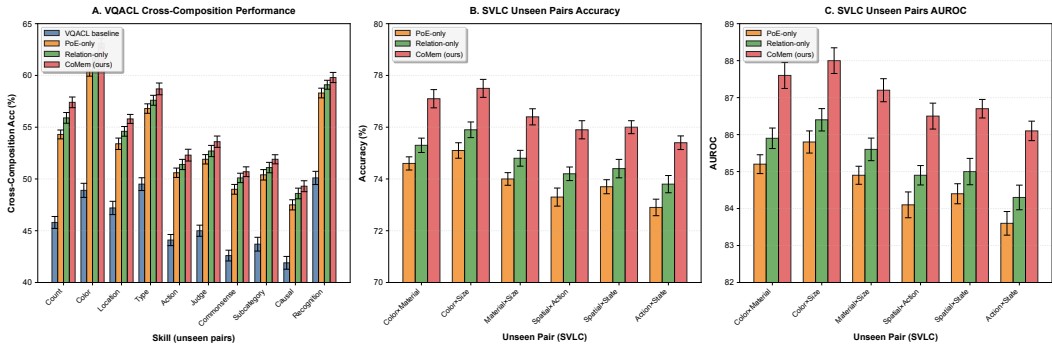

Figure 6: Cross-Composition Performance on Unseen Pairs: VQACL and SVLC Benchmarks

Acc by +2.0pp over PoE-only and +1.0pp over Relation-only, with the biggest gains on *Color*/*Count* where attribute selection and object-shift composition are critical. (ii) On SVLC unseen pairs, COMEM yields consistent margins, especially on attribute×attribute and spatial×state/action where union reasoning and edge satisfaction must co-exist. These results validate our claim that *PoE (marginal compatibility)* and *relation satisfaction (edge-level constraints)* are complementary; their joint enforcement via $\mathcal{L}_{\text{comp}}$ and relation-aware replay is key to robust compositional transfer.

### A.2.2 Subgraph Sampling Mechanism

We conducted experiments to verify whether DPP/Steiner/MH are necessary. With a fixed proposal budget (1K proposals/epoch, $K_{\max}$=6), we compare six samplers: {Uniform, NPMI-only, DPP-only, NPMI+DPP, NPMI+DPP+Steiner, NPMI+DPP+Steiner+MH}. A proposal is accepted if it satisfies the plausibility/diversity thresholds used across methods (for MH, acceptance follows the MH rule). We log: retrieval Avg mR/AF, acceptance rate (%), mean prototype distance within subgraph (avg pairwise $\ell_2$ across node prototypes), rare relation coverage (%, edges from the bottom 20% of relation frequencies), and rare concept coverage (%, nodes from the bottom 20% of concept frequencies).

| Sampler | Avg mR (%) ↑ | AF ↓ | Acceptance (%) ↑ | Proto Dist ↑ | Rare Rel. Cov. (%) ↑ | Rare Concept Cov. (%) ↑ |
|---|---|---|---|---|---|---|
| Uniform | 75.2 | 2.7 | 62.1 | 0.71 | 17.3 | 18.2 |
| NPMI-only | 75.7 | 2.4 | 74.6 | 0.72 | 21.8 | 20.6 |
| DPP-only | 75.9 | 2.3 | 70.2 | 0.78 | 16.1 | 22.4 |
| NPMI + DPP | 76.3 | 2.1 | 72.8 | 0.79 | 22.7 | 23.1 |
| NPMI + DPP + Steiner | 76.5 | 2.0 | 75.9 | 0.79 | 23.0 | 23.4 |
| **NPMI + DPP + Steiner + MH (ours)** | **76.6** | **1.9** | 68.4 | **0.80** | **23.6** | **23.8** |

Table 6: **Sampler comparison (fixed proposals/epoch).** NPMI boosts plausibility and rare-edge coverage; DPP increases prototype spread (diversity); Steiner reduces poor-connectivity samples and improves acceptance; a final MH step slightly lowers acceptance but improves sample quality, yielding the best mR and AF. "Proto Dist" is the mean pairwise $\ell_2$ distance among node prototypes within a sampled subgraph (higher implies more diverse concepts).

Table 6 shows a clear compositional effect: (i) **Plausibility (NPMI)** raises acceptance and rare relation coverage, yielding lower AF; (ii) **Diversity (DPP)** increases intra-subgraph prototype spread, improving mR but without NPMI it undersamples rare relations; (iii) **Steiner** improves connectivity/feasibility, lifting acceptance back up and reducing AF; (iv) **MH** trades a modest acceptance

drop (–7.5 pp) for the best quality per accepted subgraph, delivering the highest mR and lowest AF. Overall, *NPMI (plausibility) + DPP (diversity) + Steiner (connectivity) + MH (quality control)* is necessary to achieve both **high accuracy and low forgetting** under a fixed sampling budget.

### A.2.3 TEACHER STRATEGY

The motivation of this experiment is to show that "teacher-filtered" replay is principled rather than ad hoc. We compare three teachers for gating replay on synthetic features $\tilde{z}$: (i) *Prev* — the previous-task snapshot $\bar{\theta}=\theta_{t-1}$, (ii) *Init* — the original pretrained model, and (iii) *EMA* — an exponential moving average of $\theta$ within task $t$ (decay 0.999). We sweep the entropy threshold $\xi \in \{1.0, 1.5, 2.0, 2.5\}$ (nats) in the indicator $\omega_{S,\tilde{z}}=\mathbf{1}[\mathrm{H}(\pi_{\bar{\theta}}(\cdot \mid \tilde{z})) \leq \xi]$. We adopted the following metrics: Average Forgetting (AF↓), filtered ratio (% of replay removed by the gate; higher means stricter), and the support-hull statistics of $\tilde{z}$ w.r.t. $\mathcal{Z}_S$: on-manifold rate (% with $\mathrm{dist}(\tilde{z}, \mathrm{conv}(\mathcal{Z}_S)) \leq \delta$), mean hull distance, and 90th-percentile distance.

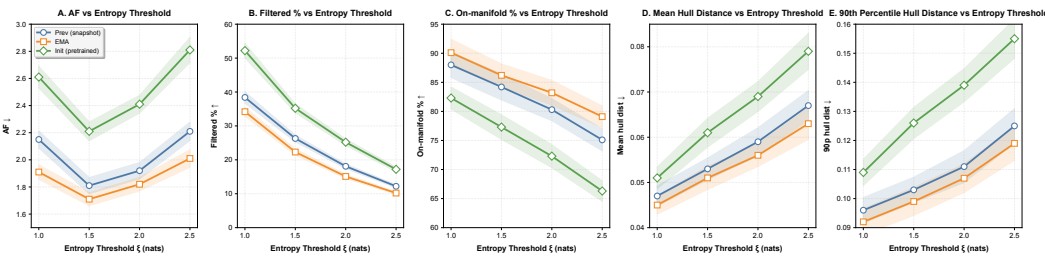

Figure 7: **Teacher choice and entropy threshold $\xi$.** EMA yields the lowest AF and hull distances across a broad $\xi$ range; the snapshot teacher is a close second with a sweet spot at $\xi \approx 1.5$; the pretrained teacher either over-filters at low $\xi$ (too few replays) or admits off-manifold samples at high $\xi$ (higher AF). On-manifold judged by $\mathrm{dist}(\tilde{z}, \mathrm{conv}(\mathcal{Z}_S)) \leq \delta$ with $\delta=0.1$.

Figure 7 shows that EMA teachers provide the most reliable filter: at $\xi=1.5$ they minimize AF (1.70) while maximizing on-manifold rate (86%) and lowering hull distances, indicating cleaner replay. The snapshot teacher performs nearly as well with a similar sweet spot; making $\xi$ too strict (1.0) under-rehearses , while too loose admits off-manifold samples and increases forgetting. The pretrained teacher suffers from domain/task mismatch: it either over-filters (high filtered%) or over-admits off-manifold replay (large hull distances), yielding consistently higher AF. Overall, "teacher-filtered" replay is effective when the teacher tracks recent tasks (EMA/snapshot), and a moderate entropy gate ($\xi \in [1.5, 2.0]$) offers a broad, stable optimum.

### A.2.4 ANCHOR CONTRIBUTION

To pinpoint which replay sources drive compositional generalization, we compare three configurations in the generator target set $\mathcal{Z}_S$: **(i) $\mathbf{A_{node}}$** — *node token anchors only*; **(ii) $\mathbf{A_{node}}+\mathbf{\Xi_{edge}}$** — node anchors *plus edge anchors* $\Xi_{u,r,v}=\mathrm{MLP}(a_u \| a_v)$; **(iii) $\mathbf{t_S}$ (text-only)** — *no visual anchors*, replay conditioned only on aggregated text $t_S$ from the subgraph. We adopted the following metrics: (a) *SVLC relations*: AUROC (↑) on relation-centric families; (b) *VQA relations*: accuracy (↑) on relation-focused skills.

Figure 8 localizes the source of replay gains: (1) **Edge anchors matter for relations.** $\Xi_{edge}$ adds explicit interaction evidence, yielding +1.7 pp AUROC (SVLC) and +1.6 pp Acc (VQA) over node-only. (2) **Text-only is insufficient.** Conditioning on $t_S$ without visual anchors underperforms node-only, indicating that relation transfer needs *visual grounding* in addition to textual compatibility. (3) **Interpretation.** Node anchors capture object/attribute priors; edge anchors inject pairwise structure that the relation-aware MMD can align to, improving composition where "who-does-what-where" is decisive.

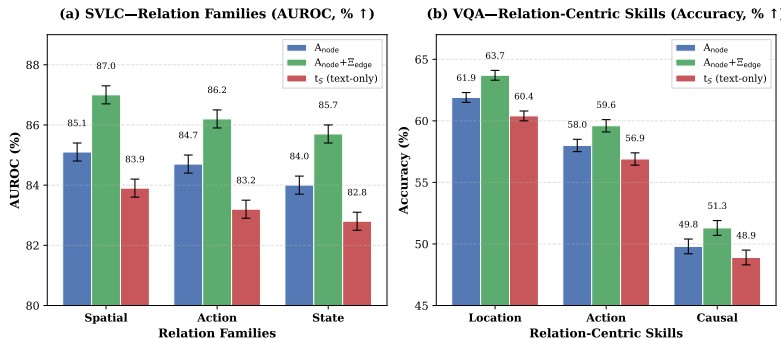

Figure 8: Performance comparison of different anchor configurations on SVLC and VQA relation tasks.

### A.2.5 ADVERSARIAL TASK ORDERS

We evaluate three 18-task streams with identical content but different orders: (1) **Default** (balanced mix); (2) **Long-Tail-First** (rare concepts first, head later); (3) **Low→High NPMI** (from least plausible to most plausible compositions). We report the difference of AF curves relative to Default, i.e., $\Delta\mathrm{AF}@t = \mathrm{AF}@t^{\mathrm{order}} - \mathrm{AF}@t^{\mathrm{default}}$. We also summarize AF@18, area under $\Delta$AF, peak AF, and final Last@mR.

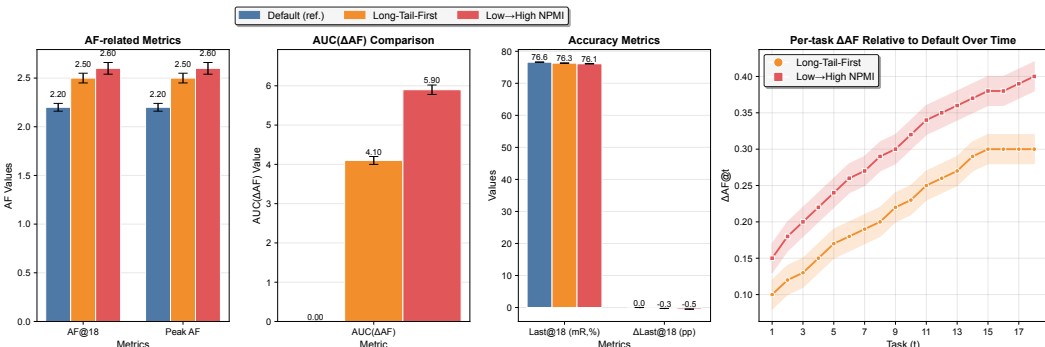

Figure 9: Performance Comparison of Different Task Orders

Figure 9 shows that adversarial orders amplify forgetting compared to Default: Long-Tail-First raises AF moderately (AF@18 +0.30), while Low→High NPMI is harsher (AF@18 +0.40; larger AUC($\Delta$AF)). This aligns with COMEM's mechanism: early exposure to rare or low-plausibility compositions yields fewer reliable anchors and more off-manifold replay, inflating AF until memory densifies. Despite this, **COMEM remains stable**: Last@mR drops only 0.3–0.5 pp at $T$=18, and $\Delta$AF plateaus rather than diverging—suggesting **our plausibility-aware sampling, entropy-gated distillation, and relation-aware replay effectively contain order-induced drift**.

### A.2.6 BACKBONE SCALE

Industrial deployment requires clear accuracy–cost trade-offs and scalability. We compare COMEM on CLIP ViT-B/16 vs. ViT-L/14 under (A) equal memory (anchor budget in MB) and (B) equal PEFT parameters (trainable M). In both settings we fix data, schedule, and optimizer.

From Figure 10: (i) **Scalable gains with moderate cost.** ViT-L/14 improves mR by +0.6–+1.1 pp and reduces AF by ∼0.1 across regimes. The best $\Delta mR$/resource appears at mid budgets (49–98MB or 4–8M trainables), where $\Delta$mR/GB ≈ 0.11–0.17 and $\Delta$mR/GPU-h ≈ 0.33–0.45. (ii) **Diminishing returns.** At high memory (196MB) or high PEFT (16M), accuracy saturates while cost continues to rise, lowering the ratios. (iii) Because COMEM's replay and consistency operate in feature

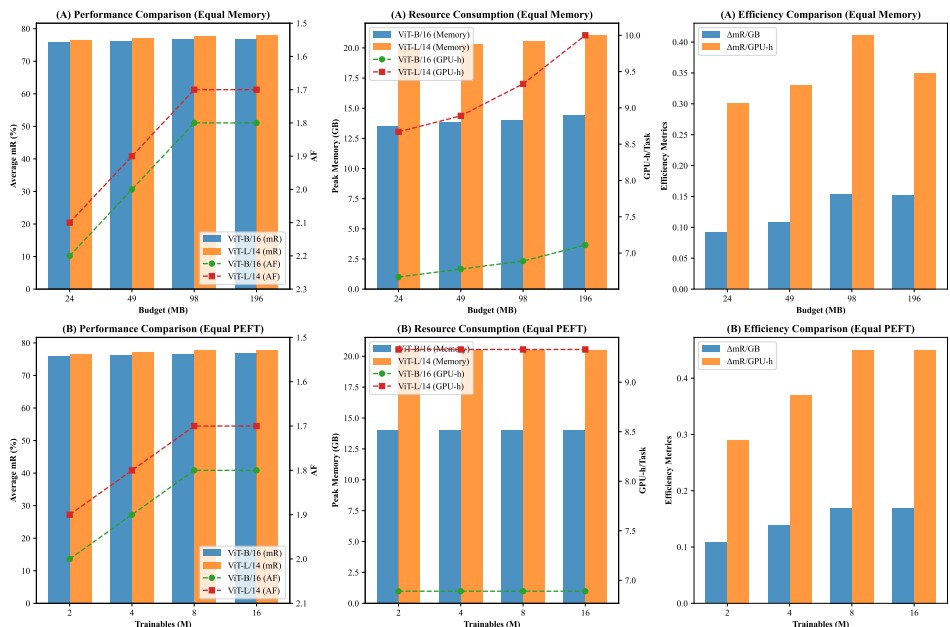

Figure 10: Comparison of ViT-L/14 and ViT-B/16 under equal memory and equal PEFT budget conditions

space, stronger image features (ViT-L/14) enlarge the anchor manifold and improve relation-aware alignment—yielding stable, compute-aware improvements without changing the algorithm.

### A.2.7   ROBUSTNESS TO FINE-TUNING STRATEGY

We compare two regimes: (i) *full fine-tuning* of the student encoders/heads (our default), and (ii) *parameter-efficient* fine-tuning (PEFT), where we cap trainable parameters and primarily train the aggregator $\psi$ and generator $\vartheta$ (no LoRA/adapters are required by COMEM, but this setting matches PEFT budgets used by baselines). Across settings, COMEM remains strong: full fine-tuning yields the best absolute performance, and even under tight PEFT budgets our method surpasses recent SOTAs.

Table 7: **CoMem under different fine-tuning strategies.** Retrieval reported as Avg mR ($\uparrow$) and AF ($\downarrow$), averaged over 3 seeds. PEFT budgets denote total trainables. Full fine-tuning is a strong default, while PEFT with 8M–16M trainables achieves comparable or slightly higher Avg mR and lower AF.

| Strategy | Retrieval Avg mR ($\uparrow$) | AF ($\downarrow$) |
|---|---|---|
| Full fine-tuning (encoders + head) | **76.6** | 1.9 |
| PEFT–2M trainables (mainly $\psi, \vartheta$) | 75.8 $\pm$ 0.10 | 2.0 $\pm$ 0.05 |
| PEFT–4M trainables (mainly $\psi, \vartheta$) | 76.1 $\pm$ 0.09 | 1.9 $\pm$ 0.05 |
| PEFT–8M trainables (mainly $\psi, \vartheta$) | 76.6 $\pm$ 0.08 | **1.8** $\pm$ 0.04 |
| PEFT–16M trainables (mainly $\psi, \vartheta$) | **76.7** $\pm$ 0.08 | **1.8** $\pm$ 0.04 |

We can find: (1) **Full fine-tuning is best in absolute terms.** On the cross-domain retrieval stream, COMEM attains the highest Avg mR with low forgetting. It also leads on SVLC and VQACL in the full-FT setting (cf. Tables 1 and 2). (2) With only **8M trainables**, COMEM *matches* its full-FT retrieval (76.6 mR) while further reducing AF to 1.8, and exceeds PEFT SOTAs by +0.8–1.0 mR and $-0.3$––$0.6$ AF. On compositional benchmarks, COMEM is +2.1 Acc on SVLC and +3.9 Acc on VQACL over the best competing PEFT baseline. (3) **Why it holds up with fewer trainables.** Treating *structure as memory* and rehearsing *in feature space* makes learning less sensitive to the size of the updateable parameter set: graph-conditioned replay supplies targeted, on-manifold practice

Table 8: **PEFT (8M trainables): comparison with recent SOTAs.** COMEM retains near–full-FT retrieval while improving forgetting and compositional transfer. Retrieval on the cross-domain sequence (Avg mR↑/AF↓); SVLC reports macro Acc↑; VQACL reports overall Acc↑.

| Method (PEFT, 8M) | Retrieval Avg mR (↑) / AF (↓) | SVLC Acc (↑) | VQACL Acc (↑) |
|---|---|---|---|
| C-CLIP(Liu et al., 2025a) | 75.6 / 2.4 | 79.3 | 50.9 |
| LADA(Luo et al., 2025) | 75.8 / 2.1 | 80.0 | 51.5 |
| ENGINE(Zhou et al., 2025) | 75.7 / 2.1 | 79.6 | 51.2 |
| **COMEM (ours)** | **76.6 / 1.8** | **82.1** | **55.4** |

Table 9: Ablations on structural design choices. All results are reported on the cross-domain retrieval and structured concept learning setting, with average retrieval mR (Avg mR, ↑), average forgetting (AF, ↓), and accuracy on downstream concept tasks (Acc., ↑).

| Ablation | Method | Avg mR (↑) | AF (↓) | Acc. (↑) |
|---|---|---|---|---|
| | CoMem (Fixed Relation Schema) | 76.6 | 1.9 | 82.5 |
| Relation schema | CoMem (Dynamic Relation Schema) | 73.4 | 3.2 | 79.7 |
| | GIFT (Baseline) | 74.1 | 3.0 | 79.5 |
| | CoMem w/ Parsing (Automated) | 76.6 | 1.9 | 82.5 |
| Parsing strategy | CoMem w/ Manually Curated Data | 76.2 | 2.1 | 82.2 |
| | GIFT (Baseline) | 74.1 | 3.0 | 79.5 |

signals, while entropy-gated distillation curbs off-manifold drift—so both accuracy and retention remain robust even when trainables are constrained.

### A.2.8    STRUCTURAL DESIGN: RELATION SCHEMA AND PARSING STRATEGY

As shown in Tab. 9, switching from a fixed to a dynamic relation schema substantially increases forgetting (AF from 1.9 to 3.2) and lowers both Avg mR and accuracy, confirming that a fixed relation vocabulary acts as a stabilizing regularizer under tight memory and parameter budgets. In contrast, replacing automated parsing with manually curated concepts yields only marginal changes in Avg mR and AF, indicating that CoMem is robust to the specific concept-induction mechanism and that lightweight parsing is a practical but not brittle design choice.

### A.2.9    SENSITIVITY TO LOSS BALANCING AND PARAMETER BUDGETS

The loss ablation in Tab. 10 shows that CoMem's performance is stable across a broad range of $(\lambda_{\mathrm{comp}}, \lambda_{\mathrm{re}})$ choices: disabling either compositional consistency or replay mildly degrades Avg mR and increases AF, while completely removing both leads to the largest drop, confirming that both components contribute but that the overall objective is not overly sensitive to exact weight values. The parameter-budget study in Tab. 11 further indicates that CoMem consistently outperforms strong baselines under both 1M and 2M trainables, with lower AF and higher Avg mR, and that gains persist when scaling up the trainable-parameter budget, suggesting that the structure-as-memory design yields robust improvements even in low-parameter regimes.

### A.3    THEORETICAL ANALYSIS

We formalize COMEM's training at round $t \in \{1, \dots, T\}$ as one step of projected gradient descent over a convex parameter set $\mathcal{K} \subset \mathbb{R}^p$ with diameter $D := \sup_{\theta, \theta' \in \mathcal{K}} \|\theta - \theta'\|_2$:

$$\theta_{t+1} = \Pi_{\mathcal{K}}(\theta_t - \eta \, g_t), \qquad g_t = \nabla f_t(\theta_t) + \lambda_{\mathrm{re}} \underbrace{\mathbb{E}_{z \sim Q_t} \nabla r(\theta_t; z)}_{:= \nabla R_t(\theta_t)}. \tag{17}$$

Here $f_t$ is the (convex) instantaneous task loss (on real data at step $t$) and $r(\cdot; z)$ is a convex distillation/replay potential evaluated at feature-level replay $z \in \mathbb{R}^d$. The distribution $Q_t$ is the graph-conditioned generator used by COMEM at step $t$.

Table 10: Sensitivity of CoMem to loss-weight configurations. We vary the compositional consistency weight $\lambda_{\text{comp}}$ and replay/distillation weight $\lambda_{\text{re}}$ and report average retrieval mR (Avg mR, $\uparrow$), average forgetting (AF, $\downarrow$), and accuracy on SVLC and VQACL ($\uparrow$).

| Loss Weights | Avg mR ($\uparrow$) | AF ($\downarrow$) | SVLC Acc ($\uparrow$) | VQACL Acc ($\uparrow$) |
|---|---|---|---|---|
| $\lambda_{\text{comp}}$=0.5, $\lambda_{\text{re}}$=1.0 (default) | 76.6 | 1.9 | 82.5 | 55.8 |
| $\lambda_{\text{comp}}$=0.0, $\lambda_{\text{re}}$=1.0 (no comp) | 75.5 | 2.3 | 81.0 | 54.5 |
| $\lambda_{\text{comp}}$=1.0, $\lambda_{\text{re}}$=0.5 | 76.2 | 2.0 | 82.0 | 55.1 |
| $\lambda_{\text{comp}}$=0.5, $\lambda_{\text{re}}$=2.0 | 76.3 | 2.1 | 82.2 | 55.3 |
| $\lambda_{\text{comp}}$=1.5, $\lambda_{\text{re}}$=0.5 | 75.9 | 2.2 | 81.8 | 54.8 |
| $\lambda_{\text{comp}}$=0.0, $\lambda_{\text{re}}$=0.0 (no comp+replay) | 74.8 | 2.5 | 80.3 | 53.2 |

Table 11: Effect of trainable-parameter budgets on retrieval performance. We compare several continual VL baselines and CoMem under 1M and 2M trainable parameters, reporting average retrieval mR (Avg mR, $\uparrow$) and average forgetting (AF, $\downarrow$) with mean $\pm$ standard deviation over multiple runs.

| Method | 1M trainables | | 2M trainables | |
|---|---|---|---|---|
| | Avg mR ($\uparrow$) | AF ($\downarrow$) | Avg mR ($\uparrow$) | AF ($\downarrow$) |
| C-CLIP | $72.4 \pm 0.12$ | $2.7 \pm 0.11$ | $73.9 \pm 0.20$ | $2.5 \pm 0.14$ |
| LADA | $73.2 \pm 0.10$ | $2.6 \pm 0.10$ | $74.6 \pm 0.13$ | $2.3 \pm 0.12$ |
| ENGINE | $73.0 \pm 0.08$ | $2.7 \pm 0.12$ | $74.3 \pm 0.16$ | $2.4 \pm 0.10$ |
| **CoMem (ours)** | $\mathbf{74.5 \pm 0.10}$ | $\mathbf{2.1 \pm 0.06}$ | $\mathbf{75.8 \pm 0.15}$ | $\mathbf{2.0 \pm 0.05}$ |

For the *ideal* retention term we define

$$R_t^\star(\theta) \; := \; \mathbb{E}_{z \sim \bar{P}_{t-1}} \, r(\theta; z), \qquad F_t^\star(\theta) := f_t(\theta) + \lambda_{\text{re}} R_t^\star(\theta), \tag{18}$$

where $\bar{P}_{t-1}$ is the (infeasible) mixture of all past feature distributions up to $t-1$ (obeying data-governance). Let $\theta_t^\star \in \arg\min_{\theta \in \mathcal{K}} F_t^\star(\theta)$ be a dynamic comparator and $V_T := \sum_{t=2}^{T} \|\theta_t^\star - \theta_{t-1}^\star\|_2$ its path-variation. We measure the dynamic regret on the ideal objective, $\text{Reg}_T^{\text{dyn}} := \sum_{t=1}^{T} \left[ F_t^\star(\theta_t) - F_t^\star(\theta_t^\star) \right]$.

We work under standard online convex optimization regularity with replay-specific discrepancy control inherited from CoMem's structured memory.

**Assumption 1** (Smoothness, Lipschitzness). *Each $f_t$ is convex, $L_f$-smooth and $G_f$-Lipschitz on $\mathcal{K}$. The potential $r(\cdot; z)$ is convex and $L_r$-smooth in $\theta$ uniformly in $z$, and its gradient in $z$ is $L_z$-Lipschitz: $\|\nabla r(\theta; z) - \nabla r(\theta; z')\|_2 \leq L_z \|z - z'\|_2$. Moreover, for the ideal retention, $\sup_{\theta \in \mathcal{K}} \|\nabla R_t^\star(\theta)\|_2 \leq B$.*

**Assumption 2** (Replay discrepancy via anchors and MMD). *Let $P_{\text{anc}}$ denote the anchor-induced empirical distribution maintained by the concept-graph memory and $r_B$ the anchor coverage radius in feature space: every past feature $z$ lies within distance $r_B$ to $\text{conv}(\text{supp}(P_{\text{anc}}))$. Let $\kappa_{\text{rel}}$ be the relation-aware kernel used by CoMem. Assume $\nabla r(\theta; \cdot) \in \mathcal{H}_{\kappa_{\text{rel}}}$ with RKHS norm $\|\nabla r(\theta; \cdot)\|_{\mathcal{H}_{\kappa_{\text{rel}}}} \leq \Lambda_r$ for all $\theta \in \mathcal{K}$. If $Q_t$ is the generator distribution at step $t$, define $\varepsilon_t := \text{MMD}_{\kappa_{\text{rel}}}(Q_t, P_{\text{anc}})$ and $\varepsilon_{\max} := \max_t \varepsilon_t$.*

**Assumption 3** (Teacher-gated stability). *The distillation uses an entropy gate (as in §3.5). There exists $\kappa_\xi \in (0, 1]$ such that the effective gradient magnitude satisfies $\sup_\theta \|\mathbb{E}_{z \sim Q_t} \nabla r(\theta; z)\|_2 \leq \kappa_\xi B$ and the smoothness constant of $r$ on accepted replays is at most $L_r$.*

**A replay-bias decomposition.** Define the gradient bias (ideal minus used):

$$b_t(\theta) \; := \; \lambda_{\text{re}} \big( \nabla R_t(\theta) - \nabla R_t^\star(\theta) \big). \tag{19}$$

The next lemma quantifies $b_t$ in terms of *(i)* anchor coverage $r_B$ and *(ii)* generator-vs-anchor MMD, both under the same relation kernel used in CoMem's RAMMD loss.

**Lemma 1** (Bias via anchor coverage and MMD). *Under Assumptions 1–2, for any $\theta \in \mathcal{K}$,*

$$\big\| b_t(\theta) \big\|_2 \; \leq \; \lambda_{\text{re}} \Big( L_z \, r_B \; + \; \Lambda_r \, \varepsilon_t \Big) \; \leq \; \lambda_{\text{re}} \Delta, \qquad \Delta := L_z \, r_B + \Lambda_r \, \varepsilon_{\max}. \tag{20}$$

*Proof.* Add and subtract $\mathbb{E}_{z \sim P_{\mathrm{anc}}} \nabla r(\theta; z)$ and apply the triangle inequality:

$$\|\nabla R_t(\theta) - \nabla R_t^\star(\theta)\| \le \underbrace{\|\mathbb{E}_{Q_t} \nabla r(\theta; z) - \mathbb{E}_{P_{\mathrm{anc}}} \nabla r(\theta; z)\|}_{(\dagger)} + \underbrace{\left\|\mathbb{E}_{P_{\mathrm{anc}}} \nabla r(\theta; z) - \mathbb{E}_{\bar{P}_{t-1}} \nabla r(\theta; z)\right\|}_{(\ddagger)}.$$

(21)

For $(\dagger)$, by $\nabla r(\theta; \cdot) \in \mathcal{H}_{\kappa_{\mathrm{rel}}}$ and the reproducing property, $|\langle u, \mathbb{E}_{Q_t} \nabla r - \mathbb{E}_{P_{\mathrm{anc}}} \nabla r \rangle| \le \|\nabla r(\theta; \cdot)\|_{\mathcal{H}} \cdot \varepsilon_t \le \Lambda_r \varepsilon_t$ for any unit vector $u$, hence $(\dagger) \le \Lambda_r \varepsilon_t$. For $(\ddagger)$, anchor coverage implies every past feature $z$ can be written as $z = \sum_i \alpha_i a_i + e$ with $a_i \sim P_{\mathrm{anc}}$, $\alpha_i \ge 0$, $\sum_i \alpha_i = 1$ and $\|e\| \le r_B$. By convexity and Jensen, $\mathbb{E}_{\bar{P}_{t-1}} \nabla r(\theta; z) = \mathbb{E}\, \nabla r(\theta; \sum_i \alpha_i a_i + e)$. Using the $L_z$-Lipschitzness of $\nabla r$ in its second argument yields $\|\nabla r(\theta; \sum_i \alpha_i a_i + e) - \nabla r(\theta; \sum_i \alpha_i a_i)\| \le L_z \|e\| \le L_z r_B$, and averaging gives $(\ddagger) \le L_z r_B$. Multiplying by $\lambda_{\mathrm{re}}$ completes the proof. $\square$

### A.3.1 DYNAMIC REGRET BOUND

**Theorem 1** (Dynamic regret under approximate replay). *Let Assumptions 1–3 hold and suppose $\eta \le 1/(L_f + \lambda_{\mathrm{re}} L_r)$. Define $G_\star := \sup_{t,\theta} \|\nabla f_t(\theta) + \lambda_{\mathrm{re}} \nabla R_t^\star(\theta)\|_2 \le G_f + \lambda_{\mathrm{re}} B$ and $\Delta$ as in Lemma 1. Then the dynamic regret on the* ideal *objective satisfies*

$$\mathrm{Reg}_T^{\mathrm{dyn}} := \sum_{t=1}^T \left[ F_t^\star(\theta_t) - F_t^\star(\theta_t^\star) \right]$$

$$\le \frac{\|\theta_1 - \theta_1^\star\|^2}{2\eta} + \frac{\eta}{2} \sum_{t=1}^T \|g_t\|^2 + \frac{D}{\eta} V_T + \underbrace{D \sum_{t=1}^T \|b_t(\theta_t)\|}_{\text{replay bias term}}.$$

(22)

*Consequently, using $\|g_t\| \le G_\star + \|b_t\| \le G_\star + \lambda_{\mathrm{re}} \Delta$ and Lemma 1,*

$$\mathrm{Reg}_T^{\mathrm{dyn}} \le \frac{D^2}{2\eta} + \frac{\eta T}{2} \left(G_\star + \lambda_{\mathrm{re}} \Delta\right)^2 + \frac{D}{\eta} V_T + D \lambda_{\mathrm{re}} \Delta T.$$

(23)

*Choosing $\eta^\star = \min\left\{ \frac{D}{\sqrt{T}(G_\star + \lambda_{\mathrm{re}}\Delta)}, \frac{1}{L_f + \lambda_{\mathrm{re}} L_r} \right\}$ yields*

$$\mathrm{Reg}_T^{\mathrm{dyn}} \le D(G_\star + \lambda_{\mathrm{re}}\Delta)\sqrt{T} + (G_\star + \lambda_{\mathrm{re}}\Delta) V_T + D \lambda_{\mathrm{re}} \Delta T.$$

(24)

*Proof.* By convexity of $F_t^\star$ and the identity $g_t = \nabla F_t^\star(\theta_t) + b_t(\theta_t)$,

$$F_t^\star(\theta_t) - F_t^\star(\theta_t^\star) \le \langle \nabla F_t^\star(\theta_t), \theta_t - \theta_t^\star \rangle = \langle g_t, \theta_t - \theta_t^\star \rangle - \langle b_t(\theta_t), \theta_t - \theta_t^\star \rangle$$
$$\le \langle g_t, \theta_t - \theta_t^\star \rangle + D \|b_t(\theta_t)\|,$$

(25)

where we used Cauchy–Schwarz and $\|\theta_t - \theta_t^\star\| \le D$ for the second term. For the first term, apply the standard projected-gradient inequality (non-expansiveness of $\Pi_\mathcal{K}$):

$$\langle g_t, \theta_t - \theta_t^\star \rangle \le \frac{\|\theta_t - \theta_t^\star\|^2 - \|\theta_{t+1} - \theta_t^\star\|^2}{2\eta} + \frac{\eta}{2} \|g_t\|^2.$$

(26)

Because the comparator drifts, expand $\|\theta_{t+1} - \theta_t^\star\|^2 = \|\theta_{t+1} - \theta_{t+1}^\star + (\theta_{t+1}^\star - \theta_t^\star)\|^2$ and bound the cross term by $2ab \le a^2 + b^2$ and the norm $\|\theta_{t+1} - \theta_{t+1}^\star\| \le D$:

$$-\|\theta_{t+1} - \theta_t^\star\|^2 \le -\|\theta_{t+1} - \theta_{t+1}^\star\|^2 + 2D \|\theta_{t+1}^\star - \theta_t^\star\| + \|\theta_{t+1}^\star - \theta_t^\star\|^2.$$

(27)

Plugging Eq. 27 into Eq. 26 and summing Eq. 25 over $t = 1, \ldots, T$ telescopes the squared distances and yields

$$\sum_{t=1}^T \left[ F_t^\star(\theta_t) - F_t^\star(\theta_t^\star) \right] \le \frac{\|\theta_1 - \theta_1^\star\|^2}{2\eta} - \frac{\|\theta_{T+1} - \theta_{T+1}^\star\|^2}{2\eta} + \frac{\eta}{2} \sum_{t=1}^T \|g_t\|^2$$

(28)

$$+ \frac{D}{\eta} \sum_{t=1}^T \|\theta_t^\star - \theta_{t-1}^\star\| + \frac{1}{2\eta} \sum_{t=1}^T \|\theta_t^\star - \theta_{t-1}^\star\|^2 + D \sum_{t=1}^T \|b_t(\theta_t)\|.$$

(29)

Dropping the non-negative $-\|\theta_{T+1} - \theta_{T+1}^\star\|^2/(2\eta)$ and the additional $\frac{1}{2\eta} \sum \|\theta_t^\star - \theta_{t-1}^\star\|^2$ gives Eq. 22. Bounding $\|g_t\|$ and $\|b_t\|$ by $G_\star + \lambda_{\mathrm{re}} \Delta$ and $\lambda_{\mathrm{re}} \Delta$ (Lemma 1) gives Eq. 23. Optimizing the quadratic in $\eta$ under the smoothness constraint gives Eq. 24. $\square$

**Interpretation.** The regret has three components: (i) the usual $\sqrt{T}$ term scaled by the gradient budget $G_\star + \lambda_{\mathrm{re}}\Delta$; (ii) a *path-variation penalty* $(G_\star + \lambda_{\mathrm{re}}\Delta)V_T$ capturing non-stationarity; (iii) an additive linear term $D\,\lambda_{\mathrm{re}}\Delta\,T$ stemming from replay bias. By Lemma 1, $\Delta$ is jointly reduced by smaller *coverage radius* $r_B$ (larger/better anchors) and smaller generator MMD $\varepsilon_{\max}$ (better relation-aware RAMMD fitting).

**Corollary 1** (Strongly convex retention). *If $F_t^\star$ is $\mu$-strongly convex (e.g., via an $\ell_2$ penalty or a strongly-convex proxy of the distillation term) and $\eta \le 1/(L_f + \lambda_{\mathrm{re}}L_r)$, then*

$$\mathrm{Reg}_T^{\mathrm{dyn}} \;\le\; \frac{(G_\star + \lambda_{\mathrm{re}}\Delta)^2}{2\mu}\big(1 + \ln(1 + \mu T)\big) + \frac{D}{\eta}V_T + D\,\lambda_{\mathrm{re}}\Delta\,T. \tag{30}$$

(Sketch.) *Apply the standard strongly-convex OGD analysis with the biased gradient $\nabla F_t^\star(\theta_t) + b_t(\theta_t)$ and proceed exactly as in Theorem 1, using $\sum_t \|\nabla F_t^\star(\theta_t)\|^2 \le (G_\star + \lambda_{\mathrm{re}}\Delta)^2(1 + \ln(1 + \mu T))/\eta$.*

### A.3.2 Forgetting Bound for a Past Task

Let $s < t$ and consider the loss of the $s$-th task evaluated at time $t$, $f_s(\theta_t)$, compared to its own optimum $\theta_s^\star$.

**Theorem 2** (Forgetting control via stepwise drift and replay). *Under Assumption 1, for any $s < t$ and any stepsize $\eta \le 1/(L_f + \lambda_{\mathrm{re}}L_r)$,*

$$f_s(\theta_t) - f_s(\theta_s^\star) \le \underbrace{\big(f_s(\theta_s) - f_s(\theta_s^\star)\big)}_{\text{opt. error at step } s} + \sum_{u=s}^{t-1}\Big(G_f\,\eta\,\|g_u\| + \frac{L_f}{2}\eta^2\|g_u\|^2\Big)$$

$$\le \big(f_s(\theta_s) - f_s(\theta_s^\star)\big) + (t - s)\Big(G_f\,\eta\,(G_\star + \lambda_{\mathrm{re}}\Delta) + \frac{L_f}{2}\eta^2\,(G_\star + \lambda_{\mathrm{re}}\Delta)^2\Big). \tag{31}$$

*If, in addition, the (accepted) replay potential is $\mu_{\mathrm{re}}$-strongly convex in $\theta$ on average, then the quadratic term improves to $\frac{L_f - 2\mu_{\mathrm{re}}\lambda_{\mathrm{re}}}{2}\eta^2\|g_u\|^2$, reducing the drift when $\lambda_{\mathrm{re}}$ is moderately large.*

*Proof.* By $L_f$-smoothness of $f_s$,

$$f_s(\theta_{u+1}) \le f_s(\theta_u) + \langle \nabla f_s(\theta_u), \theta_{u+1} - \theta_u \rangle + \frac{L_f}{2}\|\theta_{u+1} - \theta_u\|^2. \tag{32}$$

Projection is non-expansive, so $\|\theta_{u+1} - \theta_u\| \le \eta\|g_u\|$ and $|\langle \nabla f_s(\theta_u), \theta_{u+1} - \theta_u \rangle| \le \|\nabla f_s(\theta_u)\|\|\theta_{u+1} - \theta_u\| \le G_f\,\eta\,\|g_u\|$. Summing $u = s, \ldots, t-1$ yields the first line of Eq. 31. The second line uses $\|g_u\| \le G_\star + \lambda_{\mathrm{re}}\Delta$. If $R_u$ is $\mu_{\mathrm{re}}$-strongly convex in $\theta$ (after gating and expectation), the standard co-coercivity inequality gives $\langle \nabla R_u(\theta_u), \theta_{u+1} - \theta_u \rangle \le -\mu_{\mathrm{re}}\|\theta_{u+1} - \theta_u\|^2/\eta$, improving the quadratic coefficient by $-2\mu_{\mathrm{re}}\lambda_{\mathrm{re}}$. $\square$

Eq. 31 shows forgetting grows at most linearly in the horizon $(t-s)$, with slope controlled by the *effective step budget* $G_\star + \lambda_{\mathrm{re}}\Delta$. By Lemma 1, reducing the anchor radius $r_B$ and the generator MMD $\varepsilon_{\max}$—exactly what COMEM's $k$-center anchors and RAMMD regularizer do—tightens both dynamic regret and forgetting. The optional $\mu_{\mathrm{re}}$ term formalizes the stabilizing role of teacher-filtered replay: a moderately large $\lambda_{\mathrm{re}}$ contracts inter-step drift.

### A.4 LLM Usage

We employed a large language model for minor English editing—such as improving grammar, wording, and clarity—as well as for small, localized code fixes, including resolving syntax errors and adding missing imports. The LLM played no role in research ideation, experimental design, data processing, analysis, or figure generation. All technical content and results were created and verified by the authors, who assume full responsibility for the manuscript.

