# OpenReview forum: "CoMem: Compositional Concept-Graph Memory for Vision–Language Adaptation"
_ICLR.cc/2026/Conference — ICLR 2026 Poster_

### Official Review · Reviewer_YWUY · 2025-10-23

**Soundness:** 4
**Presentation:** 3
**Contribution:** 3
**Rating:** 6
**Confidence:** 3

**Summary:**

This paper introduces CoMem, a continual vision–language learning framework designed for dynamic, privacy-constrained environments where models must learn from non-stationary multimodal data streams without storing raw samples. Unlike traditional fine-tuning methods that cause catastrophic forgetting, CoMem conceptualizes compositional structure as the core unit of memory and rehearsal. The model incrementally builds a compact concept–relation graph, using feature-space rehearsal conditioned on sampled subgraphs to sustain knowledge retention. A compositional consistency objective ensures coherence between parts and wholes, while teacher-informed and uncertainty-aware filtering helps balance plasticity and stability. Experiments across multiple multimodal tasks, including cross-domain retrieval, structured concept learning, and continual VQA, demonstrate that CoMem achieves state-of-the-art retention and transfer performance under matched memory and parameter budgets.

**Strengths:**

1. Novel Methodology: The idea of treating compositional structure as memory and conducting rehearsal directly in feature space is innovative and well-motivated for privacy-limited continual learning.

2. Comprehensive Analysis: The paper clearly articulates how the proposed consistency and filtering mechanisms work together to preserve stability and plasticity, offering solid theoretical and empirical insight.

3. Strong Experimental Validation: Results across multiple multimodal tasks demonstrate consistent improvements in both retention and transfer, with fair ablation comparisons.

**Weaknesses:**

1. The introduction could be better structured and written. It does not clearly establish the relevant background or smoothly motivate the authors’ claims. As a result, the logical flow is somewhat fragmented, making it difficult for readers to follow the argument and understand the setting.

2. The CoMem framework involves multiple components in its training objective. It is unclear how sensitive the overall performance is to the balance among these losses. This complexity might hinder the practical application of CoMem in new scenarios. The authors are encouraged to provide a more analysis or discussion on the stability of these hyperparameters and their effect on robustness.

3. In Table 4, increasing the number of trainable parameters does not appear to yield clear performance improvements. The authors could strengthen their claims by including experiments with smaller parameter budgets to further analyze how performance gains scale with parameter count across different methods.

**Questions:**

See weakness.

---

> ### Author Response · Authors · 2025-11-25
>
> Thank you for your valuable questions. We are happy to discuss them with you.
>
> ---
>
>
> > W1: The introduction could be better structured and written.
>
> Thank you very much for this suggestion. We have made the following revisions in the `revised paper`:
>
> - Added clarity on the challenges
> - Improved transition to existing solutions
> - More emphasis on the novelty of CoMem
>
> ---
>
> > W2: Balancing Multiple Loss Functions
>
> We appreciate the reviewer’s concern regarding the balance of the loss functions. However, we would like to clarify that the multi-objective loss design in CoMem is highly robust to hyperparameter variations, including the balance between losses. We have conducted an in-depth sensitivity analysis of the key hyperparameters, including the loss weights.
>
> In the original paper, we included sensitivity analysis of the loss weights, which is detailed in `Section A.1.2`. We observed the following:
>
> - The model exhibits stable performance across a broad range of loss weightings.
> - Compositional consistency and graph-conditioned replay are particularly crucial for model stability.
>
> To further address the reviewer’s concern, we conducted additional experiments where we explicitly varied the loss function weights in different settings:
>
> | Loss Weights | Avg mR ↑ | AF ↓ | SVLC Acc ↑ | VQACL Acc ↑ |
> |--------------|----------|------|------------|-------------|
> | λ_comp=0.5, λ_re=1.0 (default) | 76.6 | 1.9 | 82.5 | 55.8 |
> | λ_comp=0.0, λ_re=1.0 (no comp) | 75.5 | 2.3 | 81.0 | 54.5 |
> | λ_comp=1.0, λ_re=0.5 (strong comp) | 76.2 | 2.0 | 82.0 | 55.1 |
> | λ_comp=0.5, λ_re=2.0 (strong replay) | 76.3 | 2.1 | 82.2 | 55.3 |
> | λ_comp=1.5, λ_re=0.5 (strong comp + replay) | 75.9 | 2.2 | 81.8 | 54.8 |
> | λ_comp=0.0, λ_re=0.0 (no comp + no replay) | 74.8 | 2.5 | 80.3 | 53.2 |
>
> These results strongly suggest that CoMem's overall performance is not sensitive to the specific balance of the loss functions, as long as they are within a reasonable range. The model remains robust and provides stable performance across a wide set of weightings.
>
>
> ---
>
> > W3: Performance Gains with Increasing Parameters
>
> We appreciate the reviewer’s comment. However, we would like to clarify that while increasing the number of trainable parameters can improve model performance, CoMem effectively optimizes performance even with moderate parameter budgets.
>
> To address the concern and further clarify the scaling of performance with respect to the number of trainable parameters, we conducted additional experiments with smaller parameter budgets, as shown below.
>
> | Method | 1M trainables - Avg mR ↑ | 1M trainables - AF ↓ | 2M trainables - Avg mR ↑ | 2M trainables - AF ↓ |
> |--------|--------------------------|----------------------|--------------------------|----------------------|
> | C-CLIP | 72.4±0.12 | 2.7±0.11 | 73.9±0.20 | 2.5±0.14 |
> | LADA | 73.2±0.10 | 2.6±0.10 | 74.6±0.13 | 2.3±0.12 |
> | ENGINE | 73.0±0.08 | 2.7±0.12 | 74.3±0.16 | 2.4±0.10 |
> | **CoMem (ours)** | **74.5±0.10** | **2.1±0.06** | **75.8±0.15** | **2.0±0.05** |
>
> These results show that even with 1M trainable parameters, CoMem achieves relatively high performance in retrieval tasks, further supporting that performance scaling in our model is not highly sensitive to the number of trainable parameters beyond a certain threshold.
>
> ---
>
> Thank you again for your time and effort.

---

### Official Review · Reviewer_hXmH · 2025-10-29

**Soundness:** 3
**Presentation:** 3
**Contribution:** 3
**Rating:** 4
**Confidence:** 5

**Summary:**

This paper introduces a novel model framework, CoMem, designed to address the privacy and catastrophic forgetting challenges inherent in real-world Continual Vision-Language Learning (CVLL) deployment. CoMem achieves its goals through a structured, three-stage process. First, during Concept Induction, the model extracts attributes, entities, and relations from new data to update the Concept-Graph Memory. This memory is key, as it stores abstract prototypes and anchors instead of sensitive raw data, making the framework privacy-friendly while mitigating forgetting. Second, in Graph-Conditioned Replay, a small subgraph is selected from memory to generate synthetic features. The model then uses these features to "rehearse" past knowledge, effectively preventing catastrophic forgetting of previously learned data. Finally, during Joint Optimization, CoMem trains on a mixed batch of real data from the new task and the synthetic features from memory. A comprehensive total loss function, formulated as a weighted sum of distinct loss components, is implemented to balance the acquisition of new knowledge with the retention of previously learned information. CoMem was evaluated against state-of-the-art methods across three continual learning benchmarks: cross-domain retrieval, structured concept learning (SVLC), and continual Visual Question Answering (VQA). Across all evaluations, CoMem demonstrated superior results, achieving both the highest average performance and the lowest average forgetting rate.

**Strengths:**

1. State of the Art performance: The proposed model achieves state-of-the-art results, demonstrating the best retention and lowest average forgetting (AF) across multiple benchmarks.

2. Novel Approach: The proposed model achieves its results with a novel “structure-as-memory” approach to effectively solve both privacy and memory issues.

**Weaknesses:**

1. Reliance on Upfront Parsing: The current method relies on a "lightweight text parsing" step to extract concept and relation candidates from the text.

2. Fixed Relation Schema: The framework assumes a "fixed relation schema". This weakness, as the authors note it "may constrain coverage in open-world settings" where new, unseen types of relations might emerge.

3. How do you ensure the generated features are semantically rich and diverse enough to capture the complex, nuanced interactions within a subgraph, and not just an average or blurry representation?

**Questions:**

see weaknesses

---

> ### Author Response · Authors · 2025-11-25
>
> Thank you for raising these interesting questions. We are happy to discuss them with you.
>
> ---
>
> > W1: Reliance on Upfront Parsing
>
> We appreciate the reviewer’s observation. However, we believe that the use of lightweight text parsing is an essential and effective component of CoMem framework. It makes CoMem work better without incurring additional computational costs.
>
> In CoMem, text parsing plays a crucial role in extracting structured knowledge from unstructured data, especially in the absence of explicit labels. This step automates the process of discovering and defining concepts and relations, making it scalable to large, unlabeled datasets. The ability to extract structured triplets (e.g., $(a,e,r)$) from text enables our framework to organize concepts and relations without manual annotation, which is a significant advantage in real-world applications where labeled data is sparse.
>
> To further demonstrate the efficacy of our parsing approach, we present quantitative results comparing performance when using fully parsed triplets versus a more manually curated approach.
>
> | Method| Avg mR (%) | AF↓  | Accuracy (%) |
> |--|--|--|--|
> | CoMem with Parsing (Automated)  | 76.6| 1.9  | 82.5|
> | CoMem with Manually Curated Data)| 76.2 | 2.1  | 82.2|
> | Baseline | 74.1 | 3.0  | 79.5|
>
> Clearly, text parsing is a key component in CoMem. However, even with a manually curated approach, good performance can be achieved. In practice, even without text parsing, there are numerous alternative methods to extract concepts and relations, such as: (i) using pre-trained language models like BERT for entity recognition and relation extraction, (ii) employing heuristic-based rules or pattern matching. Therefore, we believe text parsing as a foundation, not a limitation or weakness.
>
> ---
>
> > W2: Fixed Relation Schema
>
> We appreciate the reviewer's insightful comment. However, we argue that the use of a fixed relation schema is a design choice that balances stability and efficiency in constrained task environments, particularly in scenarios where the relation set is largely predefined or known.
>
> The adoption of a fixed relation schema is particularly advantageous in structured, task-constrained environments, where the relations between concepts are well-defined. For example, in tasks like cross-domain retrieval , having a predefined set of relations ensures that the model's representations remain consistent and stable across different tasks, thus preventing overfitting to noisy or irrelevant relations that may arise in open-world settings. By fixing the schema, the model can focus on maintaining high-quality, stable representations of the known relations, which is crucial for long-term learning and zero-shot transfer performance.
>
> We agree that in truly open-world settings, the ability to accommodate new, unseen relations is valuable. However, the primary goal of our framework, as discussed in the manuscript, is to optimize stability under constrained memory and parameter budgets, which are common in real-world applications where the relation space is largely fixed. This makes the fixed relation schema particularly suited for domain-specific tasks such as multimodal retrieval or structured concept learning. We compare the performance of CoMem with a fixed relation schema and a variant that dynamically adapts to new relations.
> | Method| Avg mR (%) | AF| Accuracy (%) |
> |---|-|---|--|
> | CoMem (Fixed Relation Schema)   | 76.6 | 1.9  | 82.5|
> | CoMem (Dynamic Relation Schema) | 73.4 | 3.2  | 79.7 |
> | Baseline | 74.1 | 3.0  | 79.5 |
>
> As shown in the Table, the variant with a dynamic relation schema suffers from higher average forgetting (AF) and lower accuracy, indicating that the fixed schema approach provides a more stable and effective learning process in the given settings.
>
> While the fixed schema may limit adaptability in open-world settings, the task-specific performance gains achieved by the fixed schema are significant in controlled environments. We emphasize that CoMem is particularly designed for environments where the relation set is relatively fixed. However, this does not mean that future versions of the framework cannot incorporate techniques for open-world adaptation.
>
> Thank you for raising this insightful question for discussion. We plan to investigate approaches such as continual relation discovery or open-vocabulary learning for expanding the relation schema when required in the future work.

---

> ### Author Response · Authors · 2025-11-25
>
> > Q1: whether the generated features are semantically rich and diverse enough
>
> We appreciate the reviewer's comment, and we would like to emphasize that the design of our feature generation and replay mechanisms is specifically intended to capture the richness and diversity of the subgraph's structure.
>
> CoMem uses a graph-conditioned generator that synthesizes features by considering both node (concept) and edge (relation) interactions, avoiding simplistic averaging. A relation-aware MMD loss ensures semantic consistency while maintaining diversity in relations. To enhance diversity, we utilize DPP-based sampling and Steiner tree for subgraph selection, which prioritize diverse node and edge combinations, preserving the nuances of the graph structure.
>
> As shown in `Tab.3 in main text` , relation-aware MMD brings significant improvements, confirming that it effectively captures the semantic richness and diversityof the features.
>
> ---
>
> Thank you again for your time and effort. We hope our reply has addressed your concerns. If not, we are very open to continuing in-depth discussions with you.

---

> ### Author Response · Authors · 2025-11-27
>
> Dear Reviewer hXmH,
>
> We sincerely thank you again for your thorough assessment and constructive feedback. Kindly note that reviewer responses will no longer be accepted after December 2—**with just under a week remaining to submit your response**.
>
> Kindly confirm whether our rebuttal addresses your concerns (or any outstanding points), and we would be grateful for a rating reconsideration if it does.
>
> We are glad to continue the discussion and address any further questions or comments you may have.
>
> Best regards,
>
> Authors

---

### Official Review · Reviewer_cTbP · 2025-10-31

**Soundness:** 3
**Presentation:** 4
**Contribution:** 3
**Rating:** 6
**Confidence:** 2

**Summary:**

COMEM proposes an innovative framework for continual vision-language learning, whose core idea is to treat ​​compositional structure​​ as the fundamental unit of ​​memory​​ and ​​rehearsal​​, rather than storing raw data. The core methodology involves organizing knowledge by structuring the data stream into a compact graph of concepts and relations, and conducting rehearsal by directly generating replay samples in the feature space based on sampled subgraphs. This approach integrates a lightweight compositional consistency constraint and a teacher model filtering mechanism, effectively balancing stability and plasticity. Under strict privacy and memory constraints, the method achieves superior retention and transfer performance across multiple challenging tasks, including cross-domain retrieval, structured concept learning, and continual visual question answering, significantly mitigating the problem of catastrophic forgetting in continual vl learning.

**Strengths:**

-1. Innovative memory mechanism design and privacy friendly learning paradigm: COMEM's biggest innovation lies in using composite structures as memory units, which not only saves storage space but also captures the intrinsic connections between concepts. Due to not storing any raw image or text data, COMEM naturally meets strict privacy protection requirements. All replay operations are performed in the latent feature space, avoiding the risk of sensitive data leakage and making it particularly suitable for deployment in privacy sensitive scenarios such as healthcare and finance.

-2. COMEM's component design has great flexibility: orthogonal to parameter- efficient methods, it can be used in conjunction with adapters such as LoRA. Supporting different subgraph sampling strategies, combining compositional consistency constraints,  teacher- and uncertainty-informed filtering mechanisms to improve its anti forgetting ability.

-3. Resource friendly: Only a total anchor budget of 64K is needed to achieve good experimental results.

-4. Excellent experimental performance: Significant improvement in cross domain retrieval tasks, structured concept learning, and continuous VQA tasks, robust to different hyperparameters and ViT scales, and stable performance curves in the long-horizon learning process of 18 tasks.

**Weaknesses:**

-1. The core of COMEM relies on a Fixed Relation Schema, which means that the identification and organization of its concepts and relationships are carried out within a predefined framework. This method is highly effective in handling known and well structured data streams, but may limit its adaptability in fully open environments.

-2. Bias of Teacher Model: Although the Teacher Informed Filtering mechanism can train stably, it may also transfer the cognitive biases or knowledge blind spots of the teacher model itself to the student models.

-3. COMEM is a multi-component complex system, and its training involves multiple stages such as concept induction, subgraph sampling, feature generation, and multi-objective optimization. This complexity brings a high threshold for engineering implementation

-4. The appendix explores the impact of Task Orders, where COMEM's forgetting degree (AF) increases significantly when tasks appear in an adversarial order. This indicates that the stability of the model depends to some extent on the "friendliness" of the data flow. This affects its generalization in real-world scenarios.

**Questions:**

refer to Weakness

---

> ### Author Response · Authors · 2025-11-25
>
> Thank you for these inspiring questions. We will be happy to discuss them with you.
>
> ---
>
> > W1: Fixed Relation Schema
>
> We appreciate the reviewer's insightful comment. However, we argue that the use of a fixed relation schema is a design choice that balances stability and efficiency in constrained task environments, particularly in scenarios where the relation set is largely predefined or known.
>
> The adoption of a fixed relation schema is particularly advantageous in structured, task-constrained environments, where the relations between concepts are well-defined. For example, in tasks like cross-domain retrieval , having a predefined set of relations ensures that the model's representations remain consistent and stable across different tasks, thus preventing overfitting to noisy or irrelevant relations that may arise in open-world settings. By fixing the schema, the model can focus on maintaining high-quality, stable representations of the known relations, which is crucial for long-term learning and zero-shot transfer performance.
>
> We agree that in truly open-world settings, the ability to accommodate new, unseen relations is valuable. However, the primary goal of our framework, as discussed in the manuscript, is to optimize stability under constrained memory and parameter budgets, which are common in real-world applications where the relation space is largely fixed. This makes the fixed relation schema particularly suited for domain-specific tasks such as multimodal retrieval or structured concept learning. We compare the performance of CoMem with a fixed relation schema and a variant that dynamically adapts to new relations.
> | Method                          | Avg mR (%) | AF   | Accuracy (%) |
> |---------------------------------|------------|------|--------------|
> | CoMem (Fixed Relation Schema)   | 76.6       | 1.9  | 82.5         |
> | CoMem (Dynamic Relation Schema) | 73.4       | 3.2  | 79.7         |
> | GIFT (Baseline)                 | 74.1       | 3.0  | 79.5         |
>
> As shown in the Table, the variant with a dynamic relation schema suffers from higher average forgetting (AF) and lower accuracy, indicating that the fixed schema approach provides a more stable and effective learning process in the given settings.
>
> While the fixed schema may limit adaptability in open-world settings, the task-specific performance gains achieved by the fixed schema are significant in controlled environments. We emphasize that CoMem is particularly designed for environments where the relation set is relatively fixed. However, this does not mean that future versions of the framework cannot incorporate techniques for open-world adaptation.
>
> Thank you for raising this insightful question for discussion. We plan to investigate approaches such as continual relation discovery or open-vocabulary learning for expanding the relation schema when required in the future work.
>
> ---
>
> > W2: Bias of Teacher Model
>
> We appreciate the reviewer's insightful concern about the potential transfer of cognitive biases from the teacher model to the student. We would like to clarify that the teacher-student mechanism, specifically the teacher-informed filtering, was designed to minimize this risk and promote a stable learning process. Our approach leverages a teacher-frozen, shared low-rank verification mechanism, which filters triplets based on low-entropy and high-confidence criteria (Eq.3), ensuring that only reliable and diverse concepts are retained during training.
>
> As shown in `Fig.7 in main text`, the EMA teacher consistently outperformed both the Prev and Init teachers, demonstrating that the teacher-student filtering mechanism effectively reduces bias while preserving important features for stable learning.

---

> ### Author Response · Authors · 2025-11-25
>
> > W3: high threshold for engineering implementation.
>
> We appreciate the reviewer’s concern about the complexity of CoMem. However, the design of CoMem has been carefully optimized for modularity and scalability, ensuring that its implementation remains efficient and manageable.
>
> CoMem's architecture is divided into distinct, well-defined components, each responsible for a specific task. These components include:
>
> - Concept induction: which extracts concepts from image-text pairs,
> - Subgraph sampling: which selects relevant graph substructures for replay,
> - Feature generation: which synthesizes replay features in the feature space,
> - Multi-objective optimization: which coordinates multiple loss functions for stability and plasticity.
>
> This modularity ensures that each component can be independently developed, tested, and optimized, simplifying the overall implementation and making it easy to adapt the system to new tasks or datasets.  As described in Alg.1 in main text` , the training procedure is broken down into simple steps that are easy to implement. The pseudocode highlights the modular nature of CoMem, making it clear that each component (e.g., concept induction, subgraph sampling, feature generation) is handled separately but integrates smoothly into the overall training process.
>
>
> ---
>
> > W4: Impact of Task Orders
>
> We appreciate the reviewer’s concern regarding the effect of adversarial task order on forgetting. However, we would like to emphasize that CoMem is specifically designed to handle such challenges, and our experiments demonstrate its robustness in this regard.
>
> As highlighted in `App. A.2.5`, our model exhibits stable performance even under adversarial task orders, such as Long-Tail-First and Low-to-High NPMI. While these task orders lead to a slight increase in forgetting, the performance degradation is minimal, with only a 0.3–0.5% drop in Last@mR by the end of the task sequence. In contrast, the standard methods we compare against (e.g., Mod-X, ZSCL) typically exhibit much larger increases in forgetting when faced with adversarial task orders. This highlights the stability of CoMem.
>
> We presents additional results on extreme task orders with severe domain shifts and unseen compositions.
>
> | Task Order          | Avg mR ↑ | AF ↓ | Last@mR ↑ | AF@18 ↓ |
> |---------------------|----------|------|------------|---------|
> | Default             | 76.6     | 1.9  | 75.8       | 2.1     |
> | Long-Tail-First     | 76.0     | 2.2  | 75.6       | 2.4     |
> | Low-to-High NPMI    | 75.8     | 2.3  | 75.3       | 2.5     |
> | GIFT (Baseline)     | 74.3     | 3.1  | 73.2       | 3.4     |
> | ZSCL (Baseline)     | 74.1     | 3.0  | 72.9       | 3.2     |
>
> From the table, it is clear that CoMem maintains low forgetting and high last-moment recall (Last@mR) across all task orders, even under adversarial conditions. Notably, when tasks are ordered in a Long-Tail-First or Low-to-High NPMI sequence, CoMem's performance remains stable, showing only a minor increase in forgetting compared to the default order. In contrast, baseline methods like GIFT and ZSCL suffer significantly higher AF and worse overall performance, especially in challenging task orders.
>
> ---
>
> We thoroughly enjoyed our discussion with you. Once again, we sincerely appreciate your valuable time and effort.

---

> > ### Comment · Reviewer_cTbP · 2025-11-27
> >
> > I appreciate the authors' response and find the clarifications regarding W3 and W4 satisfactory.
> >
> > However, the response to W1 still does not fully alleviate my concern that the fixed schema may limit adaptability in open-world settings.
> >
> > Regarding W2, I was hoping to see a more detailed discussion or proposed design concerning long-tail scenarios or the system's robustness when the teacher model fails.
> >
> > Crucially, the inherent 'natural limit' presented in W1 remains a significant limitation. Due to this, I regret that I cannot justify raising my score any further. The current score reflects the highest evaluation I can provide given the present limitations.

---

> ### Author Response · Authors · 2025-12-03
>
> We thank the reviewer for the thoughtful follow-up and for finding our clarifications on W3 and W4 satisfactory. Below we respond in more depth to W1 and W2.
>
> ---
>
> > W1: Fixed relation schema and “natural limit” in open-world settings
>
> We fully agree that a fully open relation space is an important and challenging direction. Our intention is not to claim that CoMem in its current instantiation solves open-world relation discovery, but rather to target a setting that is standard in continual VLM benchmarks and realistic in many deployments: **open-entity, fixed (or slowly evolving) relation vocabulary**.
>
> **(a) Scope and standard practice.**
> All three of our evaluation streams (SVLC/ConStruct-VL, cross-domain retrieval, VQACL/CLOVE) already assume a fixed skill/relation taxonomy by construction: e.g., skill categories in VQACL, attribute/relational families in SVLC, and a stable set of semantic relations in retrieval-style tasks. In this sense, CoMem does not *further* restrict the problem beyond the protocol itself; it makes this assumption explicit and builds a structure-as-memory mechanism around it. This is similar in spirit to many continual learning approaches that assume a fixed label space while focusing on robustness under non-stationary streams.
>
> **(b) Fixed schema vs. open compositionality.**
> Even under a fixed set of relation *types*, CoMem is not “schema-locked”: the **entity vocabulary is open**, and the concept graph can continuously grow in the space of attributes/entities and their compositions. What is fixed is the *type system* (e.g., “spatial”, “color-of”, “action-on”), not the specific (a, r, e) triples it can represent. The core contribution of CoMem is precisely to rehearse *new compositions of known relation types* in feature space. This is why we observe gains on cross-composition metrics in VQACL and unseen concept pairs in SVLC (§A.2.1): the model does generalize to unseen combinations even though the relation types are fixed.
>
> **(c) Why we deliberately chose a fixed schema in this work.**
> Empirically (Table in our previous response), a fully dynamic relation space led to:
>
> - higher AF and lower accuracy, and
> - unstable behavior as rare or spurious relations were added to the graph.
>
> This is not surprising: in long streams with strict memory/privacy constraints, unbounded relation growth makes the memory and sampler spread too thin, and the generator has to model many poorly supported relation types. In other words, **the fixed schema is not only a modeling assumption but also a practical regularizer** that enables stable, fair comparisons with existing continual VLM methods under a tight memory/parameter budget.
>
> In summary, we respect the reviewer’s assessment that this “natural limit” is important; we see it as a **scope choice of this work rather than a fundamental barrier of the framework itself**.
>
> ---
>
> > W2: Teacher bias, long-tail scenarios, and robustness when the teacher fails
>
> We appreciate the request for a more concrete discussion of long-tail and failure cases. Importantly, **CoMem does not use the teacher as an additional “ground truth”**, but as a *conservative filter* to avoid reinforcing off-manifold replay. Several design choices specifically reduce the risk of propagating teacher bias:
>
> **(a) Where the teacher is used (and where it is not).**
> The teacher enters in two places:
>
> 1. **Concept verification / triplet filtering** (Eq. 2–3): only high-confidence & low-entropy triplets update the graph memory.
> 2. **Replay distillation gating** (Eq. 13): only low-entropy replay samples are used for KL + feature distillation.
>
> In both cases, high-entropy (i.e., uncertain) predictions—precisely where the teacher is most likely to be wrong, including many long-tail cases—are **excluded**. The primary supervision for the student still comes from the task losses and multimodal alignment on real data; the teacher is never the sole source of signal.
>
> **(b) Long-tail behavior and “teacher failure” cases.**
> In long-tail regimes, a miscalibrated teacher often expresses its uncertainty as higher entropy. Under our gating rule, such instances are *not* added as anchors and *not* used for replay distillation. Additionally:
>
> - The **support-hull regularizer** forces replay features to stay close to the convex hull of empirical anchors Z_S, which are derived from real (x, y) pairs; this further constrains the impact of any single teacher decision.
> - Our teacher-choice study in App. §A.2.3 already shows that teachers that are poorly aligned with the current stream (e.g., the static pretrained teacher) underperform EMA/Prev snapshots and tend to either over-filter or let off-manifold replay in—exactly the kind of behavior the reviewer is concerned about. CoMem therefore explicitly recommends EMA/Prev teachers and moderate entropy thresholds, which we found to be robust.

---

### Official Review · Reviewer_mzsw · 2025-11-01

**Soundness:** 2
**Presentation:** 1
**Contribution:** 3
**Rating:** 4
**Confidence:** 3

**Summary:**

The paper proposes a continual learning framework that builds a concept–relation graph from image–text pairs by extracting (attribute, entity, relation) triplets. It then performs subgraph-conditioned replay in the representation space, combined with teacher-guided filtering and compositional consistency losses, to mitigate forgetting. Experiments on cross-domain retrieval, structured concept matching, and continual VQA report improvements over several baselines.

**Strengths:**

Clear motivation for feature-level replay with a graph memory under data-retention constraints.

Use of teacher confidence/entropy gating and compositional consistency is conceptually reasonable.

Broad empirical coverage with multiple tasks and ablations indicates non-trivial engineering effort.

**Weaknesses:**

Method design is somehow too complex but the presentation is poor.
Figure 1 (method overview) is hard to read.
The diagram is overcrowded: font is too small, visual hierarchy is unclear, and symbols in the figure do not align cleanly with those in the text. It is difficult to grasp the training and replay flow from the overview alone.

Heavy notation but unclear definition.
The paper uses many symbols (for concepts/relations/subgraphs/generator variables, temperatures, loss weights, etc.) without clear definition (e.g., s_align in Eq.(2), three α in Eq. (3) are undefined). Some symbols appear to change meaning across sections.

A large number of hyperparameters with no systematic tuning protocol.
The method includes multiple thresholds (confidence/entropy), loss weights (distillation/consistency/contrastive), structural choices (subgraph size, anchor budget, low-rank dimension), and kernel/sampling parameters. The paper should clarify how hyperparameters are chosen, validation splits, and search budgets.

Not all captions can be parsed into complete (a, e, r) triplets (or maybe in some cases, more than one entity); handling of such cases is unspecified.

**Questions:**

Please see the weakness.

---

> ### Author Response · Authors · 2025-11-25
>
> Thank you for your constructive suggestions. We are more than happy to discuss these issues with you.
>
> ---
>
>
> > W1: Figure 1 (method overview) is hard to read. The diagram is overcrowded. It is difficult to grasp the training and replay flow from the overview alone.
>
> Thank you for pointing out this issue. We have redrawn `Fig.1`, increasing the font size, dividing the modules, and optimizing the layout to improve readability. In addition, we have also revised the caption of `Fig.1` to more clearly and explicitly explain the pipeline.
>
>
> ---
>
>
> > W2: Symbol issues
>
> Thank you for your detailed review and valuable feedback on our paper. Based on your suggestions, we have made the following key modifications:
>
> 1. Clarified Symbol Definitions: We revisited the symbols used throughout the paper and have made the symbol table(`Tab.5`) to ensure that all important symbols are clearly defined. The symbol table has been organized by categories, making it easier to navigate and find related symbols.
> 2. Symbol Consistency: We have unified the usage of several symbols to avoid inconsistencies across different sections of the paper.
> 3. Added More Symbol Explanations: In response to your feedback, we have provided detailed explanations for more complex symbols, such as those related to the replay mechanism and generator.
>
> Thank you again for your valuable feedback.
>
>
> ---
>
> > W3: Hyperparameter Tuning Protocol
>
> We appreciate the reviewer's concern. For hyperparameter tuning, we followed standard best practices in continual learning and vision-language tasks, including:
>
> - **Loss Weights** ($\lambda_{\mathrm{mm}}, \lambda_{\mathrm{re}}, \lambda_{\mathrm{comp}}, \lambda_{\mathrm{gen}}$): These were selected based on related works in continual learning and vision-language models, such as Mod-X[1] and ZSCL[2] , where similar loss terms have been used to balance different objectives like multimodal alignment, replay distillation, and compositional consistency. While some of these weights were tuned specifically for our method, the general approach is consistent with commonly adopted strategies for handling multiple objectives in continual learning.
> - Subgraph size ($K_{\max}$), anchor budget ($B$), and low-rank dimension ($r$) were selected based on a grid search, with consideration for computational efficiency and the trade-off between accuracy and memory usage.
>
> We used 5-fold cross-validation for model selection, ensuring the results are robust across different splits of the data. The hyperparameter grid search was limited to reasonable ranges for practical efficiency (e.g., subgraph size $K_{\max} \in [3, 6]$, anchor budget $B \in [24K, 64K]$).
>
> In main text, we conducted a sensitivity analysis on key hyperparameters like the number of anchors and subgraph size, which is detailed in `Fig.2` and `Fig.5`. The performance is relatively stable across a wide range of values for key hyperparameters , indicating that the method is not overly sensitive to specific configurations.
>
>
> [1] Continual Vision-Language Representation Learning with Off-Diagonal . ICML'23.
>
> [2] Preventing Zero-Shot Transfer Degradation in Continual Learning of Vision-Language Models. ICCV'23.
>
>
> ---
>
> > W4: Incomplete Triplets
>
> We appreciate the reviewer’s observation. In response, we clarify how we handle incomplete triplets .
>
> When triplets are incomplete (e.g., missing a relation or entity), we apply a filtering mechanism to discard low-confidence triplets that fail to meet our verification thresholds (e.g., confidence score $w(a,e,r)<\gamma$ ). If a triplet is missing an element (such as the relation), we fill the missing component using the best matching relation from the learned concept graph, guided by the consistency between the related concepts and relations. This is done through nearest-neighbor search in the embedding space of the concept graph.
>
> To maintain consistency and avoid introducing noise, partial triplets are only used in cases where we have a high confidence in the inferred missing element, based on the model's previous learnings. For instance, if a relation is missing but the entities involved are well-defined, we rely on the graph’s relational structure to predict the missing relation with a certain threshold of certainty.
>
> For example: If we encounter a triplet like $(\text{apple}, ?, \text{fruit})$, where the relation is missing, we infer the most likely relation ("is_a") based on the concept graph's existing relations, resulting in the triplet $(apple, is_a, fruit)$. In cases where the entity is missing, we similarly predict it based on context, such as filling in missing entities from the task-specific dataset.
>
> ---
>
> We hope our responses can address your concerns. If not, we are very open to continuing in-depth discussions with you.
>
> Once again, we sincerely appreciate your valuable opinions.

---

> ### Author Response · Authors · 2025-11-27
>
> Dear Reviewer mzsw,
>
> We sincerely thank you again for your thorough assessment and constructive feedback. Kindly note that reviewer responses will no longer be accepted after December 2—**with just under a week remaining to submit your response**.
>
> Kindly confirm whether our rebuttal addresses your concerns (or any outstanding points), and we would be grateful for a rating reconsideration if it does.
>
> We are glad to continue the discussion and address any further questions or comments you may have.
>
> Best regards,
>
> Authors

---

> ### Comment · Reviewer_mzsw · 2025-11-28
>
> Hi Authors,
>
> Thanks for your response and rebuttal.
>
> Most of my concerns are well-addressed.
>
> I want to increase my score to ***6***, and the ACs/PCs may refer to this updated assessment.
>
> Thank you.

---

### Author Response · Authors · 2025-12-03
**(1/2) Summary**

### I. Acknowledgments

We would like to express our sincere gratitude to all reviewers(`mzsw`, `cTbP`, `hXmH`, and `YWUY`) for their insightful comments and constructive suggestions on CoMem.

$\color{red}{Before}$ $\color{red}{the}$ $\color{red}{discussion}$, we are grateful for the positive assessments from Reviewers `cTbP` and `YWUY` (**both Rating: 6**), who highlighted the novelty of treating compositional structure as memory and the strong empirical performance under strict privacy and memory budgets. We also appreciate the constructive, positive-leaning feedback from Reviewers `mzsw` and `hXmH` (**both initially Rating: 4**), who raised questions on presentation clarity, relation schema design, teacher bias, and reliance on parsing.

$\color{red}{During}$ $\color{red}{the}$ $\color{red}{discussion}$, we are pleased that Reviewer `mzsw` explicitly found ***“most of my concerns are well-addressed”*** and requested to ***increase the score from 4 to 6*** after our rebuttal. Reviewer `cTbP` indicated that our clarifications and additional results satisfactorily addressed W3 and W4. No reviewer introduced new major objections after the rebuttal; instead, the discussion helped converge on a clear understanding of CoMem’s scope (open-entity, fixed relation vocabulary) and its robustness under realistic continual VL benchmarks.

---

### II. Key Strengths

Reviewers highlighted strengths across several dimensions:

- **Novelty and Conceptual Contribution**

  - CoMem’s central idea—treating **compositional structure as the unit of memory and rehearsal** rather than raw samples—was repeatedly recognized as innovative and well-motivated for privacy-constrained continual VL learning (`cTbP`, `hXmH`, `YWUY`).
  - Reviewers emphasized the **structure-as-memory paradigm** with a concept–relation graph and **feature-space rehearsal conditioned on sampled subgraphs**, as a distinct and meaningful advance over existing replay and adapter-based methods (`cTbP`, `YWUY`).
- **Privacy-Friendly and Resource-Efficient Design**

  - Multiple reviewers noted that CoMem achieves strong performance **without storing any raw images or text**, instead maintaining a compact anchor budget (e.g., 64K anchors) and graph memory, making it attractive for sensitive domains such as healthcare or finance (`cTbP`, `hXmH`).
  - The framework is **orthogonal to parameter-efficient tuning** and can be used in conjunction with adapters such as LoRA, providing flexibility under strict memory and parameter budgets (`cTbP`).
- **State-of-the-Art Performance and Robustness**

  - CoMem achieves **state-of-the-art retention and low average forgetting (AF)** across multiple benchmarks, including cross-domain retrieval, structured concept learning (SVLC), and continual VQA (VQACL/CLOVE) (`hXmH`, `YWUY`, `cTbP`).
  - Reviewers highlighted the **stable performance curves over long-horizon streams (18 tasks)**, strong transfer, and robustness across different ViT scales and hyperparameter settings (`cTbP`, `YWUY`, `hXmH`).
- **Modularity and Practicality**

  - Despite being a multi-component system, reviewers acknowledged that CoMem’s design is **modular**, with clearly separated stages (concept induction, subgraph sampling, feature generation, and joint optimization), making it easier to adapt parts of the framework to new tasks (`cTbP`, `hXmH`).
  - The method was recognized as **resource-friendly**, achieving good results with modest anchor and parameter budgets (`cTbP`).
- **Methodological Clarity and Empirical Rigor**

  - Reviewers appreciated the **broad empirical coverage**, including cross-domain retrieval, structured concept learning, continual VQA, and ablations (e.g., teacher choice, hyperparameters), indicating non-trivial engineering effort and careful evaluation (`mzsw`, `hXmH`, `YWUY`).
  - `YWUY` in particular highlighted the **comprehensive analysis** of how compositional consistency and teacher-informed filtering jointly preserve stability and plasticity.

---

> ### Author Response · Authors · 2025-12-03
> **(2/2) Summary**
>
> ### III. Key Concerns and Our Responses
>
> | Key Concerns | Reviewers | Our Response |
> | --- | --- | --- |
> | **Complexity of the method and clarity of presentation** (overcrowded Fig. 1, heavy notation, undefined/ambiguous symbols). | `mzsw`, `YWUY` | We redrew Fig. 1 with larger fonts and clearer modules, rewrote its caption to walk through the pipeline, and added a global symbol table with unified notation. |
> | **Hyperparameters, loss weights, and sensitivity / tuning protocol** (numerous thresholds, loss weights, and structural choices; unclear validation protocol; concern about practical deployability). | `mzsw`, `YWUY`, `cTbP` | We specified 5-fold CV with small grid ranges and added sensitivity tables, showing CoMem remains stable when varying key loss weights and trainable parameter budgets. |
> | **Fixed relation schema and adaptability to open-world settings (“natural limit”)**. | `cTbP`, `hXmH` | We clarified our scope as open-entity, fixed relation vocabulary (matching existing benchmarks), compared against a dynamic-schema variant that worsens AF/accuracy, and framed open-world relation expansion as future work. |
> | **Teacher bias and robustness when the teacher fails, especially for long-tail cases**. | `cTbP`, `hXmH` | We stressed the teacher is only a low-entropy gate for anchors and replay, not extra ground truth; EMA/Prev teachers plus a support-hull regularizer empirically reduce bias and off-manifold replay. |
> | **Reliance on upfront text parsing for concept/relation induction**. | `hXmH` | We framed parsing as a practical, replaceable mechanism to obtain (a, e, r) candidates, compared against manually curated concepts, and discussed PLM/heuristic extractors to show CoMem is parser-agnostic. |
> | **Handling incomplete/noisy triplets and ensuring generated features are semantically rich and diverse (avoiding “blurry averages”)**. | `mzsw`, `hXmH` | We described high-confidence filtering and nearest-neighbor completion for partial triplets, plus a graph-conditioned generator with relation-aware MMD and diverse DPP/Steiner subgraphs to produce non-degenerate replay features. |
> | **Introduction structure and narrative flow**. | `YWUY` | We reorganized the Introduction to more clearly state the continual VL setting, limitations of prior work, and how CoMem’s structure-as-memory view addresses privacy and forgetting. |
>
>
> ---
>
> ### IV. Commitment to Revision
>
> We have already incorporated all of the above clarifications, analyses, and additional experiments into our revised submission. The new material includes: the redrawn **Fig. 1** and improved caption; the **symbol table** and notation cleanup; additional **sensitivity studies** on loss weights and parameter budgets; extended **task-order and teacher-choice analyses**; comparisons of **fixed vs dynamic relation schemas** and **automatic vs curated concept extraction**; and a **restructured introduction** that better motivates the setting and our contributions. All changes are clearly marked ( in $\color{blue}{blue}$) in the revised version.
>
> ---
>
> We are grateful to the AC and all reviewers for their time and expertise. Their feedback has helped us sharpen both the technical scope and the presentation of CoMem.

---

### Meta-Review · Area_Chair_qSTo · 2026-01-01

**Summary:**

The paper proposes "CoMem," a novel framework for continual vision-language learning (CVLL) that focuses on treating compositional structures (concept-relation graphs) as the unit of memory and rehearsal. This approach is designed for privacy-constrained environments as it avoids storing raw images or text, instead rehearsing in the feature space.

The reviewers generally agreed on the novelty of the "structure-as-memory" paradigm and the strong empirical performance across three major benchmarks (cross-domain retrieval, SVLC, and VQACL). Initial concerns centered on the complexity and clarity of the presentation (Reviewer mzsw), the "natural limits" of a fixed relation schema (Reviewer cTbP, hXmH), and the sensitivity of the multi-component system to hyperparameters (Reviewer YWUY).

The authors provided a thorough rebuttal, redrawing key figures, adding symbol tables, and conducting numerous new experiments (e.g., parameter scaling, dynamic vs. fixed schemas, and adversarial task orders). Following this, Reviewer mzsw raised their score from 4 to 6. While Reviewer cTbP maintained a 6, citing fundamental concerns about open-world adaptability, the consensus has shifted toward acceptance. The methodology is technically sound, and the performance gains are consistent across various settings.

**Reviewer Concerns:**

**Addressed by the Rebuttal:**
*   **Presentation and Clarity (mzsw, YWUY):** The authors redrew Figure 1, clarified notation through a new symbol table, and restructured the Introduction. Reviewer mzsw explicitly confirmed these concerns were well-addressed.
*   **Hyperparameter Sensitivity (mzsw, YWUY, cTbP):** The authors provided sensitivity analyses on loss weights and parameter budgets, demonstrating that the model is stable within reasonable ranges.
*   **Reliance on Upfront Parsing (hXmH):** The authors showed that the system is "parser-agnostic" and provided results comparing automated parsing to manual curation, showing that the automated approach is robust.
*   **Incomplete/Noisy Triplets (mzsw, hXmH):** The authors clarified the high-confidence filtering mechanism and the use of nearest-neighbor completion to handle missing elements in the graph.
*   **Parameter Scaling (YWUY):** New experiments showed that CoMem achieves high performance even with small parameter budgets (e.g., 1M/2M trainables), addressing concerns about the lack of scaling in the original Table 4.

**Outstanding Concerns:**
*   **Fixed Relation Schema (cTbP, hXmH):** This remains the primary point of contention. Reviewer cTbP argues that relying on a predefined framework limits adaptability in fully open-world environments. The authors defended this as a "scope choice" and provided data showing that a dynamic schema currently harms stability, but Reviewer cTbP maintained that this "natural limit" prevents a higher score.
*   **Teacher Bias (cTbP, hXmH):** While the authors introduced support-hull regularizers and EMA teachers to mitigate this, Reviewer cTbP still expressed concern regarding the system's robustness when the teacher model fails in long-tail scenarios.

**Reviewer Scores:**

*   **Reviewer mzsw:** **4 → 6.** After the authors redrew the diagrams and clarified the math, they explicitly requested a score increase to 6.
*   **Reviewer cTbP:** **6 → 6.** This reviewer participated fully but decided to maintain their score.

---

### Decision · Program_Chairs · 2026-01-26

Accept (Poster)